

# Volatile Oxidation Products and Secondary Organosiloxane Aerosol from $D_5$ + OH at Varying OH Exposures

Hyun Gu Kang[1], Yanfang Chen[2], Jiwoo Jeong[2], Yoojin Park[3], Thomas Berkemeier[1], Hwajin Kim[2,4]
[1]Multiphase Chemistry Department, Max Planck Institute for Chemistry, 55128 Mainz, Germany
[2]Department of Environmental Health Sciences, Graduate School of Public Health, Seoul National University, 08826 Seoul,
South Korea
[3]Department of Environmental Science and Engineering, College of Engineering, Ewha Womans University, 03760 Seoul,
South Korea
[4]Institute of Health and Environment, Graduate School of Public Health, Seoul National University, 08826 Seoul, South
Korea
*Correspondence to*: Hwajin Kim (khj0116@snu.ac.kr) and Thomas Berkemeier (t.berkemeier@mpic.de)
**Abstract.** Siloxanes are composed of silicon, oxygen, and alkyl groups and are emitted from consumer chemicals. Despite
being entirely anthropogenic, siloxanes are being detected in remote regions and are ubiquitous in indoor and urban
environments. Decamethylcyclopentasiloxane ($D_5$) is one of the most common cyclic congeners, and smog chamber and
oxidation flow reactor (OFR) experiments have found $D_5$ + OH to form secondary organosiloxane aerosol (SOSiA). However,
there is uncertainty about the reaction products, and the reported SOSiA mass yields ($Y_{SOSiA}$) appear inconsistent. To quantify
small volatile oxidation products (VOP) and to consolidate the $Y_{SOSiA}$ in the literature, we performed experiments using a
Potential Aerosol Mass OFR while varying $D_5$ concentration, humidity, and OH exposure ($OH_{exp}$). We use a proton transfer
reaction time-of-flight mass spectrometer to quantify $D_5$, HCHO, and HCOOH, and detect other VOP, which we tentatively
identify as siloxanols and siloxanyl formates. We determine molar yields of HCHO and HCOOH between 52 – 211 % and 45
– 127 %, respectively. With particle size distributions measured with a scanning mobility particle sizer, we find $Y_{SOSiA}$ to be <
10 % at $OH_{exp}$ < $1.3 \times 10^{11}$ s cm$^{-3}$ and ~20 % at $OH_{exp}$ corresponding to that of the lifetime of $D_5$ at atmospheric OH
concentrations. We also find that $Y_{SOSiA}$ is dependent on both organic aerosol mass loading and $OH_{exp}$. We use a kinetic box
model of SOSiA formation and aging (aging-VBS model) to reconcile the $Y_{SOSiA}$ values found in this study and the literature.
The model uses a volatility basis set (VBS) of the primary oxidation products as well as an aging rate coefficient in the gas
phase, $k_{age,gas}$, of $2.17 \times 10^{-11}$ cm$^3$ s$^{-1}$, and an aging rate coefficient in the particle phase, $k_{age,particle}$, which is ten times smaller.
The combination of primary VBS and OH-dependent oxidative aging predicts SOSiA formation much better than a standard-
VBS parameterization that does not consider aging ($R^2$ = 0.970 vs. 0.847). The need for an ageing-dependent parameterization
to accurately model SOSiA formation shows that concepts developed for secondary organic aerosol precursors, which are able
to form low-volatile products at low $OH_{exp}$, do not necessarily apply to $D_5$ + OH. The resulting yields of HCHO and HCOOH
and the parameterization of $Y_{SOSiA}$ may be used in larger scale models to assess the implications of siloxanes on air quality.
**Keywords:** $D_5$ siloxane, organic aerosol, proton transfer reaction mass spectrometer, oxidation flow reactor, chemical kinetics





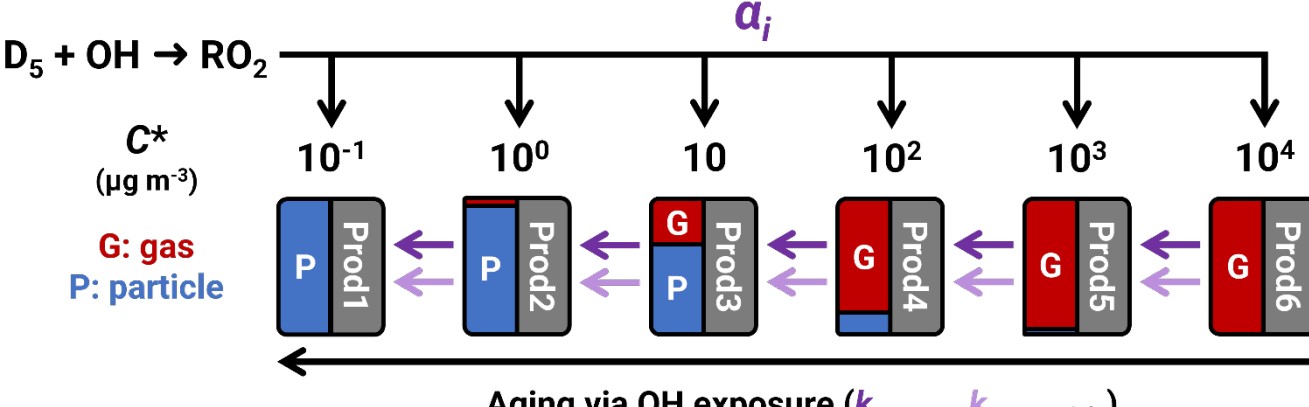

**Graphical Abstract:** Schematic of the kinetic box model.





## 1 Introduction

Organosiloxanes are molecules composed of silicon-oxygen bonds with alkyl groups on the silicons and encompass linear and cyclic species, some of which have vapor pressures on par with volatile organic compounds (VOC). Siloxanes are entirely anthropogenic pollutants (Rücker and Kümmerer, 2015) commonly used in consumer and industrial chemical products (Seltzer et al., 2021a; Gkatzelis et al., 2021) and their emissions are projected to increase in the coming decades (Tansel and Surita, 2017). Decamethylcyclopentasiloxane ($D_5$, $C_{10}H_{30}O_5Si_5$), where "D" refers to units of $(CH_3)_2SiO$, is a ubiquitous cyclosiloxane in the ambient environment.

Siloxanes can be detected in the indoor environment (Tang et al., 2015; Tran and Kannan, 2015; Arata et al., 2021; Katz et al., 2021; Kaikiti et al., 2022; Wang et al., 2022), near landfills (Schweigkofler and Niessner, 1999), and sewage treatment sites (Lee et al., 2014; Horii et al., 2019). Siloxanes are also found in outdoor urban air (Xiang et al., 2021), and organosilicon compounds have been found in varying amounts in ambient particulates in China (Lu et al., 2019; Cheng et al., 2021; Meng et al., 2021; Song et al., 2022; Xu et al., 2022) and the United States (Milani et al., 2021).

Siloxanes are suspected to be environmentally persistent or "pseudo persistent" (Howard and Muir, 2010; Xiang et al., 2021), but this long-lifetime assessment is disputed (Graiver et al., 2003; Whelan and Kim, 2021). Reaction rate coefficients of $D_5$ with atmospheric oxidants have been reported, and Atkinson (1991) found $D_5$ to be effectively unreactive with atmospheric concentrations of $O_3$ ($k_{D5+O3} < 3 \times 10^{-20}$ cm$^3$ s$^{-1}$) and $NO_3$ radicals ($k_{D5+NO3} < 3 \times 10^{-16}$ cm$^3$ s$^{-1}$) at ~298 K. While $D_5$ is reactive with OH and Cl, Alton and Browne (2020) calculated that the removal of $D_5$ by Cl radicals would only be a few percent of that by OH radicals at typical ambient oxidant concentrations.

Atkinson (1991), Safron et al. (2015), Xiao et al. (2015), Kim and Xu (2017), and Alton and Browne (2020) have measured $k_{D5+OH}$ at ~298 K to be $1.55 \times 10^{-12}$, $2.6 \times 10^{-12}$, $2.46 \times 10^{-12}$, $1.46 \times 10^{-12}$, and $2.1 \times 10^{-12}$ cm$^3$ s$^{-1}$, respectively. These measurements are summarized in Table S1. Xiao et al. (2015) derived $k_{D5+OH}$ computationally as $2.90 \times 10^{-12}$ cm$^3$ s$^{-1}$. In this paper, we use $k_{D5+OH} = 2.0 \times 10^{-12}$ cm$^3$ s$^{-1}$, which is a rounded average of the empirically determined rate coefficients. This $k_{D5+OH}$ corresponds to a $D_5$ atmospheric lifetime of ~4 days via removal by OH, assuming a daily average OH concentration ([OH]$_{avg}$) of $1.5 \times 10^6$ cm$^{-3}$.

$D_5$ is expected to suppress $O_3$ formation in urban environments. Carter et al. (1993) performed a series of chamber experiments mimicking urban air conditions and found that D5 siloxane would inhibit ozone formation by suppressing the OH radical. In contrast, formaldehyde (HCHO) is known to contribute to $O_3$ formation (Derwent et al., 1996). Fu et al. (2020) predicted the formation of HCHO as a product of $D_5$ + OH at low $NO/HO_2$ conditions using quantum chemical calculations and kinetics modelling, but an experimental yield of HCHO from $D_5$ + OH has not been reported. Atkinson (1991) proposed HCHO as a



product of the siloxane alkoxyl radical (RO) pathway, assuming an analogous mechanism to that of VOC. Sommerlade et al.
(1993) suggested that HCHO may arise from siloxane RO decomposition and from ROOH rearrangement in the presence of
acids and $H_2O$. Alton and Browne (2022) predicted HCHO as a product of $RO_2$ rearrangement in the case of $D_3$ siloxane.
Because HCHO is a secondary product, the $O_3$ formation potential of $D_5$ may differ between source and downwind locations.

Formic acid (HCOOH) is a common acid catalyst in the atmosphere (Hazra et al., 2014)  and a particle-nucleating species (Yu,
2000). Studies have identified some HCOOH sources in the atmosphere (Millet et al., 2015; Franco et al., 2021), however,
HCOOH is suspected to have unidentified anthropogenic sources in the troposphere (Millet et al., 2015; Chen et al., 2021)  as
some urban sources remain unaccounted for (le Breton et al., 2012; Yuan et al., 2015). Chandramouli and Kamens (2001)
proposed that the $RO_2$ initially formed from $D_5$ + OH makes a siloxanyl formate (D4T-OCHO, where "T" refers to $CH_3SiO$)
that reacts with $H_2O$ to a siloxanol (D4T-OH) and HCOOH. However, we are unaware of experimental HCOOH yields reported
for $D_5$ + OH.

Whelan et al. (2004) used known siloxane chemistry in a partitioning model to assess the atmospheric fate of siloxanes and
found that silanols are the predominant oxidation products. These silanols are generally water soluble and either removed from
the atmosphere via wet deposition, or undergo a pH-dependent process of hydrolysis, forming smaller and smaller silanols
(Whelan et al., 2004). Eventually, the small silanols are converted to $SiO_2$, $H_2O$, and $CO_2$ through photolytic reactions in water
or    biological    processes    in    soil    (Spivack    et    al.,    1997;    Stevens,    1998;    Graiver    et    al.,    2003).

The intermediate products between $D_5$ and those small silanols are less well studied. The intermediates may be composed of
a variety of alcohols, aldehydes, and hydroperoxides if the reaction mechanisms of $D_5$ behave in a similar manner to that of
organics, with oligomers in the condensed phase (Chen et al., 2023). However, there is evidence that siloxanes do not
necessarily follow such reaction mechanisms (Sommerlade et al., 1993; Alton and Browne, 2020, 2022), so there is a need to
understand the formation of volatile oxidation products (VOP) and secondary aerosol.

Secondary aerosol mass yield ($Y$, Eq. 1) is defined as the ratio of produced aerosol mass ($\Delta m$(SOSiA)) to reacted precursor
mass ($\Delta m$($D_5$)), which we adopt here for secondary organosiloxane aerosol (SOSiA). Reports about secondary aerosol
formation from $D_5$ siloxane are conflicting, with some experiments reporting much higher $Y_{SOSiA}$ than others. For instance, Wu
and Johnston (2017) and Janechek et al. (2019) saw maximum $Y_{SOSiA}$ of 23 % and 50 %, respectively, in their photo-oxidation
chamber and oxidation flow reactor (OFR) experiments, albeit at different OH exposures ($OH_{exp}$). Charan et al. (2022) found
a $Y_{SOSiA}$ of 158 % with their OFR at an $OH_{exp}$ of $3.2 \times 10^{12}$ s $cm^{-3}$. Avery et al. (2023) reported a wide range of $Y_{SOSiA}$ (2 – 146
%) from their PAM-OFR experiments.



$$Y_{SOSiA} = \frac{\Delta m(SOSiA)}{\Delta m(D_5)} \qquad (1)$$

In contrast, Charan et al. (2022) reported almost negligible $Y_{SOSiA}$ (< 5 %) from their chamber studies where [OH] was on the order of ~$10^6$ cm$^{-3}$, which is closer to [OH] found in ambient conditions (Peng and Jimenez, 2020). Han et al. (2022) conducted OFR experiments and found that $Y_{SOSiA}$ would be 2 % at [OH] of $4.6 \times 10^8$ cm$^{-3}$ or OH$_{exp}$ of $5.5 \times 10^{10}$ s cm$^{-3}$. The variation of $Y_{SOSiA}$ reported in the literature suggests that oxidation conditions need to be considered to accurately parameterize $Y_{SOSiA}$, especially given that $D_5$ is being considered in air quality models as a part of volatile chemical product inventories (Pennington et al., 2021; Seltzer et al., 2021a, b). In this study, we aim to develop parameterizations that reconcile the reported $Y_{SOSiA}$ for use in such air quality models.

## 2 Method and Materials

### 2.1 Experiments

The Aerodyne Research (Billerica, MA, USA) PAM-OFR (Kang et al., 2007) has a volume of 13.3 L and is made of chromated aluminum (Xu and Collins, 2021). We operated the PAM-OFR in "OFR185" mode (Peng and Jimenez, 2020), where 185 nm lamps that also emit 254 nm light (GPH436T5VH, LightSources, Orange, CT, USA) generate OH and $O_3$ with injected $H_2O$ vapor from a Nafion humidifier (FC-100-80-6MKK, Perma Pure, Lakewood, NJ, USA). There were two of these 185 nm lamps placed across from each other in clear fused quartz sleeves. The 185 nm lamps were wrapped with covers at even intervals to reduce the UV intensity so that 90 % of the lamp surface was covered. We operated the PAM-OFR at residence times ($\tau_{res}$) of 120 and 180 s with flow rates of 6.65 and 4.43 L min$^{-1}$, respectively. Additional details about the experiment setup are summarized in Fig. S1 and Sect. S1.

We used the $D_5$ siloxane trace measured from the proton transfer reaction mass spectrometer (PTR-MS) to calculate OH$_{exp}$ with Eq. (2), where $k_{D5+OH} = 2.0 \times 10^{-12}$ cm$^3$ s$^{-1}$. [$D_5$]$_0$ and [$D_5$]$_{final}$ are the $D_5$ concentrations before and after the exposure to OH.

$$OH_{exp} = -\frac{1}{k_{D5+OH}} \times ln\left(\frac{[D_5]_{final}}{[D_5]_0}\right) \qquad (2)$$

Prior to experiments, we checked the background particle and $D_5$ concentrations with the scanning mobility particle sizer (SMPS) and PTR-MS. In all experiments, the background particle number concentrations were < 10 cm$^{-3}$, and the background [$D_5$] were below the limit of detection ($3\sigma$ = 80 ppt). Then, we injected $D_5$ with a syringe pump while monitoring the PTR-MS, with major ions at $m/z$ 371 and $m/z$ 355. We performed the experiments with target [$D_5$]$_0$ of 50, 100, or 200 ppb. With these target [$D_5$]$_0$, we get external OH reactivities (OHR$_{ext}$) of 2.5 – 9.8 s$^{-1}$ at 298 K and 1 atm (Peng and Jimenez, 2020).




When the $D_5$ trace stabilized near the target $[D_5]_0$, we began the experiment by turning on the UV lamps in the PAM-OFR to
either 2.4 or 8.0 V. We waited 30 minutes for the UV lamps to stabilize and for the PAM-OFR walls to equilibrate with gaseous
species. The $Y_{SOSiA}$ (Eq. (1)) were calculated using the average SOSiA mass concentration from four SMPS cycles following
those 30 minutes. We obtained $\Delta m(D_5)$ as the difference between $[D_5]_0$ and $[D_5]_{final}$. At the end of an experiment, we turned
off the UV lamps to check the $D_5$ trace return.

To clean the PAM-OFR between experiments, we stopped the syringe pump and removed the syringe from the glass bulb
while keeping the humid air flow through into the PAM-OFR. We turned on the PAM-OFR UV lamps and connected the
outlet directly to the exhaust, until $D_5$ and particle number concentrations we below the limit of detection. We used Igor Pro 9
(Wavemetrics, Portland, OR, USA) for data post-processing and visualization.
**2.2 Instrumentation**
**2.2.1 PTR-MS**
To measure $D_5$ and VOP, we used a PTR-MS (PTR-TOF 1000, Ionicon Analytik, Innsbruck, Austria) equipped with the
extended volatility range (EVR) option (Piel et al., 2021), where the wetted inlet components and the drift tube are passivated
with a silicon coating. The PTR-MS also had ion transfer lens between the drift tube and time-of-flight mass spectrometer
(Jordan et al., 2009). An internal permeation source (PerMaSCal) emitted a steady stream of 1,3-diiodobenzene into the mass
spectrometer for mass calibration scale adjustments. Additional PTR-MS details are in Sect. S1.

To reduce $H_2O$ clusters at high humidities, we operated the PTR-MS at 137 Td ($U_{drift}$ = 600 V, Td = Townsend, 1 Td = $10^{-17}$
V cm$^2$) for quantification. The drift tube pressure and temperatures were set to 2.30 mbar and 80 °C. For the reagent ion source,
we set the $U_s$, $U_{so}$, and the $H_2O$ flow rate to 150 V, 80 V, and 6.00 sccm respectively. The ion source hollow cathode discharge
current was set to 5.0 mA. The PTR-MS drift tube was 9.6 cm long, and at 137 Td, the $(H_2O)H^+$ reaction time ($\Delta t$) was 94 µs
(de Gouw et al., 2003). We calculated the primary reagent ion signal, $(H_2O)H^+$, by multiplying the signal of its isotope,
$(H_2^{18}O)H^+$, by 500.

We used the PTR-MS data for the quantification of $D_5$ (m/z 371), HCHO (m/z 31), and HCOOH (m/z 47), where the primary
reagent ion counts were normalized to $10^6$ counts per second (ncps). For $D_5$, we used a calibration gas cylinder (Apel-Riemer
Environmental, Miami, FL, USA) containing $D_5$ to calibrate the PTR-MS. We also calculated the normalized measurement
sensitivity (ncps ppb$^{-1}$) of $D_5$, HCHO, and HCOOH using Eq. (3) adapted from de Gouw and Warneke (2007). $I_{(VOC)H^+}$ and
$I_{(H2O)H^+}$ are the ion counts of the protonated VOC and the reagent ion respectively. Additional details on the mass spectra
interpretation and quantification are in Sect. S1.5 and S3.






$$Sensitivity = \frac{\frac{I_{(VOC)H^+}}{I_{(H_2O)H^+}} \times 10^6}{[VOC]} \tag{3}$$


We tested the instrument sensitivity response with humidity by keeping the species concentrations constant while changing the sample air humidity. The sensitivity of $D_5$ at $m/z$ 371 was not heavily affected by humidity at 137 Td, and we did not correct for humidity in the $D_5$ quantification (Fig. S5). On the other hand, HCHO and HCOOH sensitivities varied with humidity, and we corrected their sensitivities as detailed in Sect. S3. Prior to experiments, we tuned the micro channel plate (MCP) to prevent signal bias against higher mass ions (Müller et al., 2014). We adjusted the MCP voltage in steps to increase the signal strength at $m/z$ 331, a PerMaSCal ion, until the relative signal increase was < 20 %.

**2.2.2 Scanning Mobility Particle Sizer**

An SMPS (Model 3938, TSI, Shoreview, MN, USA) equipped with an impactor (0.0508 cm) measured the particle mobility diameter size distribution between diameters of 14.3 to 723.4 nm. The SMPS consisted of a Model 3082 Electrostatic Classifier, a Model 3081A Differential Mobility Analyzer (DMA), a Model 3088 Soft X-ray Neutralizer, and a Model 3756 Ultrafine Condensation Particle Counter. We set the SMPS sheath flow at 3.0 L min$^{-1}$ and the aerosol flow rate at 0.3 L min$^{-1}$, and the DMA voltage ranged from 10.6 to 9921.4 V. The SMPS scanned for 150 s, followed by a 5 s retrace and 10 s purge while recording on a 3 min cycle. We referred to the manufacturer's recommendations when deciding the above SMPS settings (TSI Inc., 2012), and a sample particle size distribution from experiment 12 (Table 1) is shown in Fig. S4.

For the $Y_{SOSiA}$ calculations, we converted the SMPS integrated particle volumes into mass using a SOSiA mass density ($\rho_{SOSiA}$) of 1.07 g cm$^{-3}$ for all experiments. We obtained this $\rho_{SOSiA}$ from PAM-OFR experiments separate from the ones described here, where we weighed the masses of SOSiA collected on filters and particle volumes with the SMPS. Additional details on $\rho_{SOSiA}$ are available in Sect. S2.

**2.3 Volatility Distribution Parameterization**

Janechek et al. (2019) and Charan et al. (2022) fitted their $Y_{SOSiA}$ data to the Odum two-product model (Odum et al., 1996) and we follow the same methodology for comparison with the literature (Sect. S4). Similarly, we fit the standard volatility basis set (VBS) parameters $\alpha$ (Donahue et al., 2006) in Eq. (4) to the measured $\Delta m(SOSiA)$ using the measured $\Delta m(D_5)$, where $\alpha_i$ is the product mass yield for volatility bin $i$.

$$\Delta m(SOSiA) = \Delta m(D_5) \times \sum_{i=1}^{n} \frac{\alpha_i}{1 + \frac{C_i^*}{C_{OA}}} \tag{4}$$




In the experiments, the organosiloxane aerosol mass loading ($C_{OA}$) was equivalent to the SOSiA mass concentrations. As the
produced aerosol mass in the experiments ranged from 3.7 to ~1000 μg m$^{-3}$, we use six logarithmically spaced effective
saturation mass concentration ($C^*$) bins ranging from 0.1 to 10000 μg m$^{-3}$ at 298 K to cover the low and high-volatility products.
For reference, $D_5$ liquid has a vapor pressure of 20.4 Pa at 298 K or $C^* = 3.05 \times 10^6$ μg m$^{-3}$ (Lei et al., 2010).

As the experiments were performed for a range of $OH_{exp}$, the products between experiments may have varied due to
multigenerational aging (Zhao et al., 2015). To account for aging and parameterize $Y_{SOSiA}$ as a function of OH exposure, we
also analyse the yield data using a kinetic box model with four chemical reactions (R1–R3) written in MATLAB (MathWorks,
Natick, MA, USA).

$$D_5 + OH \rightarrow \sum \alpha_i \times prod(i) \qquad (k_{D5+OH} = 2.0\times10^{-12} \text{ cm}^3 \text{ s}^{-1}) \qquad (R1)$$
$$(1\text{-}f_i) \times prod(i) + OH \rightarrow (1\text{-}f_i) \times prod(i\text{-}1) \quad (k_{age,gas}, i = 2, \ldots, 6) \qquad (R2)$$
$$f_i \times prod(i) + OH \rightarrow f_i \times prod(i\text{-}1) \qquad (0.1 \times k_{age,gas}, i = 2, \ldots, 6) \qquad (R3)$$

Eq. (R1) describes the initial oxidation of $D_5$ and formation of $RO_2$, which immediately forms products of varying volatility
(Eq. (R2)). Here, prod(i) refers to the sum of products (gas + particle) in volatility bin $i$, which are formed with a molar
branching ratio $\alpha_i$. We assume that prod(i) have the same molecular weights (g mol$^{-1}$) as $D_5$, and so the $\alpha_i$ are equivalent to the
product mass yields at $OH_{exp} \rightarrow 0$. In the model, a fraction $f_i$ of each oxidation product partitions instantaneously from the gas
phase to the particle phase according to absorptive partitioning theory (Donahue et al., 2006) (Eq. 5).

$$f_i = \left( \frac{1}{1 + \frac{C_i^*}{C_{OA}}} \right) \qquad (5)$$

Eqs. (R2) and (R3) describe how $OH_{exp}$ causes volatility to decrease (Robinson et al., 2007). This decrease in volatility via
"bin-hopping" (Sommers et al., 2022) occurs at a rate proportional to the chemical aging rate coefficient for gaseous species
($k_{age,gas}$, cm$^3$ s$^{-1}$), with the oxidation of particle-phase products being ten times slower than that of the gas. Note that we assume
that products in the lowest-volatility bin cannot be removed from that bin and that the highest-volatility bin does not receive
product with aging ($i = 2, \ldots, 6$). The [OH] are set by dividing the experimental $OH_{exp}$ from Eq. 2 by the PAM-OFR residence
times.

We use $k_{age,gas}$ and $k_{age,particle}$ as aggregate chemical aging rate coefficients, not specific to any species or volatility bin. Studies
on chamber experiments (Robinson et al., 2007) and ambient measurements (Sommers et al., 2022) applied chemical aging
only to the gas phase as heterogeneous aging is relatively slower. However, studies have found that the high oxidant



concentrations in OFRs would appreciably oxidize OA within experiment timescales (Kessler et al., 2012; Kroll et al., 2015).
To accommodate OH uptake to the bulk phase, we follow the approach used by Zhao et al. (2015) and assume that the effective
particle-phase aging rate coefficient ($k_{age,particle}$) is equivalent to 10 % of the gas-phase aging rate coefficient ($k_{age,gas}$). The
timescales and atmospheric relevance of heterogeneous oxidation in OFRs are areas of ongoing research (Zhao et al., 2019;
Peng and Jimenez, 2020), but for now we opt to fit a single chemical aging rate coefficient to reduce dimensionality. We use
the Monte Carlo genetic algorithm (Berkemeier et al., 2017) to fit $k_{age,gas}$ and the six coefficients $\alpha_i$.

## 251 3 Results and Discussion

### 252 3.1 Volatile Organic Products (VOP)

#### 253 3.1.1 Siloxanol and Formate Ester Trends

Fig. 1 shows the PTR-MS mass spectra for experiment 12 (Table 1), where $[D_5]_0$ and $OH_{exp}$ were high. The PTR-MS signals
before and after $D_5$ is oxidized are displayed relative to the protonated $D_5$ ion at $m/z$ 371 to identify changes more easily in the
mass spectra. Using the mass spectra and species reported by Alton and Browne (2022), we attribute the indicated ions in Fig.
1 to siloxanol ($D_4T$-OH), siloxanediol ($D_3T_2$-$(OH)_2$), siloxanyl formate ($D_4T$-OCHO), and siloxanolyl formate ($D_3T_2$-OH-
OCHO). Here, "D" and "T" refers to units of $(CH_3)_2SiO$ and $CH_3SiO$ respectively. The multifunctional VOP are reported to
arise from multiple steps of oxidation (Alton and Browne, 2022).

$D_5$ siloxane loses a methyl group during the PTR, which forms a large signal at $m/z$ 355. The isotopologues of the -$CH_3$
fragment of $D_5$ overlap with fragments of VOP, which complicates the VOP identification. To separate the signal of the VOP,
we use the ratios of the $D_5$ signal and its -$CH_3$ signals prior to oxidation. The red and pink shaded areas in the inset of Fig. 1
refer to the enhancement in signal over that of the -$CH_3$ fragment of $D_5$, which we attribute to the -OH fragments of $D_4T$-OH
and $D_3T_2$-$(OH)_2$, respectively. We use the masses of the -OH fragments of the siloxanols as large alcohols dissociate during
the PTR (Brown et al., 2010).

As we did not have calibration standards to quantify these VOP, we calculate the relative molar yields of the VOP to that of
protonated $D_5$ siloxane at $m/z$ 371 to study the trends of siloxane VOP (Fig. 2). The y-axes in Fig. 2 are the relative molar
yields (ncps/ncps), which refers to the change in signal attributed to a VOP over that of $m/z$ 371. $\Delta m371$ refers to the change
in the signal at $m/z$ 371 before and after OH oxidation. In the right-side panels for each VOP in Fig. 2, the relative molar yield
of VOP decreases with increasing $OH_{exp}$ (x-axes). This decrease in VOP signal is consistent with these gaseous products
undergoing further oxidation or increased gas-particle partitioning due to higher $C_{OA}$ at higher $OH_{exp}$.





In the left-side panels for each VOP in Fig. 2, the relative signals of the VOP (y-axes) decreases with increasing $OH_{exp}$ (color
scale). Then, assuming [OH] is consistent throughout the PAM-OFR, that $D_5$ + OH is the rate-limiting step in VOP formation,
and that removal via gas-particle partitioning is negligible, we can consider a simplified $D_5$ + OH chemical mechanism, (R4)
and (R5).

$$D_5 + OH \rightarrow \sum \gamma_i \, VOP_i \qquad (k_{D5+OH} = 2.0 \times 10^{-12} \; cm^3 \; s^{-1}) \tag{R4}$$

$$VOP_i + OH \rightarrow \qquad (k_{VOPi+OH}) \tag{R5}$$


In Eq. (R4), $\gamma_i$ is the relative molar yield of a given $VOP_i$ found by extrapolating $\Delta m(VOP_i)/\Delta m371$ (y-axes in Fig. 2) to $OH_{exp}$
$\rightarrow 0$. With ordinary differential equations (ODE) from these reactions (Eq. (7) and (8)) and experimental inputs, we fit $\gamma_i$ and
the $VOP_i$ + OH rate coefficient ($k_{VOPi+OH}$, $cm^3 \; s^{-1}$). The fits are shown as black lines in the right-side panels of each VOP in
Fig. 2.

$$\frac{d[D_5]}{dt} = -k_{D5+OH}[D_5][OH] \tag{7}$$


$$\frac{d[VOP_i]}{dt} = \gamma_i k_{D5+OH}[D_5][OH] - k_{VOPi+OH}[VOP_i][OH] \tag{8}$$


The fitted $k_{VOPi+OH}$ for each VOP are on the order of $\sim 10^{-12} \; cm^3 \; s^{-1}$ (Table S7), but faster than $k_{D5+OH}$, which suggests that these
VOP have atmospheric lifetimes shorter than that of $D_5$. Alton and Browne (2022) have estimated these VOP to be volatile
with quantitative structure activity relationship models. However, there are uncertainties in those models, and the VOP may
have lower saturation mass concentrations than expected. Moreover, the chemical mechanism might be more complex than
the one outlined with the simple reactions (R4) and (R5). Consequently, we present these $k_{VOPi+OH}$ as estimates for secondary
chemistry in this simplified reaction scheme, and future work using quantitative measurements should improve the calculated
lifetimes of these intermediate $D_5$ + OH products in the atmosphere.
**3.1.2 Formaldehyde (HCHO) Yields**
As shown in Table S8 and Fig. 3, the experimental molar yields of HCHO ($Y_{HCHO}$, $\Delta HCHO/\Delta D_5$ in ppb/ppb) exceed 100 % at
low $OH_{exp}$ and decrease with higher $OH_{exp}$. We attribute the decreasing $Y_{HCHO}$ with increasing $OH_{exp}$ to HCHO removal by OH
in the PAM-OFR. HCHO has a lifetime of 0.91 days at $[OH]_{avg} = 1.5 \times 10^6 \; cm^{-3}$ (Atkinson et al., 2006) or 78 s at [OH] = 1.5
$\times 10^9 \; cm^{-3}$. In such high [OH] conditions, some HCHO is oxidized while being produced, which is consistent with the
decreasing $Y_{HCHO}$ with increasing $OH_{exp}$ (Fig. 3a1). Thus, we use the ODE from Eqs. (7) and (8) and a fixed $k_{HCHO+OH} = 8.5 \times$
$10^{-12} \; cm^3 \; s^{-1}$ at 298 K to obtain the molar yield of HCHO as $OH_{exp} \rightarrow 0$, which we denote as $\gamma_{HCHO}$.



We fit $\gamma_{HCHO}$ to be 269 % (black line in Fig. 3a2), assuming a constant [OH] in the PAM-OFR, that HCHO is rapidly formed
from $D_5$ + OH, and that HCHO removal via partitioning or reactive uptake is negligible. This $\gamma_{HCHO}$ is consistent with the
modeled yields of those for VOC used by Millet et al. (2006), who used $\gamma_{HCHO}$ from chemical models ranging from 60 – 230
% for a variety of VOC. Thus, $D_5$ has a comparable $\gamma_{HCHO}$ to that of isoprene or aromatic VOC.

Fu et al. (2020) proposed a mechanism for $D_3$ siloxane, where high $Y_{HCHO}$ is produced under low NO/HO$_2$ conditions. In that
mechanism, RO$_2$ rearrangement and RO H-shift rate coefficients become progressively faster as the $D_3$ siloxane backbone is
oxidized, and HCHO is produced at each rearrangement step. The $\gamma_{HCHO}$ exceeding 100 % in these $D_5$ experiments is consistent
with HCHO production over multiple rapid oxidation steps. The results we report suggests that a similar HCHO production
mechanism exists for $D_5$.

Mao et al. (2009) found that models under-predicted tropospheric HCHO during their aircraft campaign studying Asian
pollution outflows into the Pacific ocean. This discrepancy between the measurements and calculations was pronounced near
the surface and up to 2 km. They proposed that there is some missing OH reactivity, and that the unaccounted species would
be reactive with OH and yield HCHO when oxidized. Based on the $D_5$ experiments present here, the inclusion of siloxane
species may reduce the HCHO formation gap; Coggon et al. (2021) already noted that including volatile chemical products in
their model would increase HCHO production.

The large formation of HCHO may entail that $D_5$ siloxane could contribute to O$_3$ formation, albeit indirectly. We were unable
to observe O$_3$ enhancement due to the high concentrations of O$_3$ produced from the PAM-OFR internal chemistry itself and
the lack of NO$_x$. Given that $k_{D5+OH}$ is relatively slow compared to that of other common anthropogenic VOC, we suspect that
the oxidation of $D_5$ will occur downwind of urban sources in low-NO$_x$ conditions or in cases of air stagnation. Whether $D_5$ has
a net positive or negative effect on O$_3$ formation in these VOC/NO$_x$ scenarios needs to be assessed with models. To get a rough
estimate of O$_3$ production, we consider a case where 20 ppt of $D_5$ react with OH to form 40 ppt of HCHO, which also fully
react. This $D_5$ concentration is within the range reported by Coggon et al. (2018) in ambient urban air. The molar maximum
incremental reactivity (MIR) of HCHO under high-NO$_x$ conditions is ~20 % (Carter et al., 1995), which makes HCHO a
prominent precursor for tropospheric O$_3$. By multiplying the MIR with the HCHO reacted with OH, we can estimate an O$_3$
formation potential of 8 ppt from $D_5$ in urban air.

### 3.1.3 Formic Acid (HCOOH) Yields

We find molar yields of HCOOH ($Y_{HCOOH}$, $\Delta HCOOH/\Delta D_5$ ppb/ppb) between 45 – 127 %, as shown in Fig. 3b, although a
trend with OH$_{exp}$ is not obvious (Fig. 3b). We assume HCOOH loss via OH oxidation to be minor given the rate coefficient of
$k_{HCOOH+OH} = 4.5 \times 10^{-13}$ cm$^3$ s$^{-1}$ at 298 K (Atkinson et al., 2006), which corresponds to 17 days of OH$_{exp}$ at [OH]$_{avg}$ = 1.5 × 10$^6$
cm$^{-3}$ or an OH-oxidation lifetime of 440 s in our highest OH$_{exp}$ experiment. In addition to D$_4$T-OCHO hydrolysis, HCOOH





may have been produced by heterogeneous reactions of HCHO at the surface of the SOSiA or the OFR walls in these humid
experiments. In the atmosphere, HCOOH is presumed to form heterogeneously from HCHO and methanediol ($HOCH_2OH$) in
the presence of wet particles (Franco et al., 2021).

The $Y_{HCOOH}$ from $D_5$ + OH we report are higher than the values from isoprene + OH (Link et al., 2020) or monoterpene + OH
reported by Friedman and Farmer (2018), who quantified the $Y_{HCOOH}$ of 7 monoterpenes at varying $OH_{exp}$ without $NO_x$. The
range of $Y_{HCOOH}$ from these references is shown as shaded areas in Fig. 3b2. The $Y_{HCOOH}$ from $D_5$ is on par with the humid
isoprene ozonolysis cases reported by Link et al. (2020). Friedman and Farmer (2018) also used a PAM-OFR, but with 254
nm UV lamps in dry conditions (~1 % RH), and Link et al. (2020) used a reaction chamber, which limits a direct comparison
with our results. Nevertheless, Friedman and Farmer (2018) found $Y_{HCOOH}$ of 0.64 – 8.5 % at $OH_{exp} = 2.0 \times 10^{11}$ s cm$^{-3}$. Aside
from the different precursor VOC and mechanism, Friedman and Farmer (2018) may have encountered less heterogenous
production of HCOOH due to the dry OFR conditions. Our laboratory findings suggest that $D_5$ siloxane should be considered
as an atmospheric HCOOH source.
**3.2 SOSiA Mass Yields**
**3.2.1 Volatility Basis Set Parameterization**
The Odum two-product model does not reconcile the $Y_{SOSiA}$ in the literature in the high $C_{OA}$ range (Sect. S4), so we apply a
VBS model. Fig. 4a shows the fitted aerosol mass yield curve (black line) using a standard-VBS model (Eq. (4)), but the $Y_{SOSiA}$
(y-axis) appears to depend on both $C_{OA}$ (x-axis) and $OH_{exp}$ (color scale). To address whether accounting for the varying $OH_{exp}$
in these experiments would improve the VBS model outputs, we fit the produced SOSiA mass using a standard-VBS model
(Eq. (4)) and a kinetic model with VBS and chemical aging rate coefficients ("aging-VBS model", Eqs. (R1) – (R3)) based on
$OH_{exp}$ and $[D5]_0$ (Table 1). We fit $k_{age,gas}$ in Eq. (R2) to be $2.17 \times 10^{-11}$ cm$^3$ s$^{-1}$. The fitted VBS parameters are summarized in
Table S11.

In both the standard and aging-VBS model fits (blue and red, respectively in Fig. 4b), ~95 % of the $D_5$ + OH product mass is
in the gas phase at a $C_{OA}$ of 10 µg m$^{-3}$. The high fraction of gaseous products is consistent with low $Y_{SOSiA}$ in the lower $OH_{exp}$
experiments, whereas additional oxidation in the higher $OH_{exp}$ experiments leads to a shift towards products that partition into
the particle phase, thus increasing $Y_{SOSiA}$. Secondary organic aerosol (SOA) often exhibits a maximum yield as a function of
$OH_{exp}$, after which the yield decreases due to fragmentation becoming dominant at high $OH_{exp}$ (Isaacman-VanWertz et al.,
2018). We do not find such a maximum in the range of $OH_{exp}$ studied, which suggests that an even higher $Y_{SOSiA}$ could have
been found at higher $OH_{exp}$. Moreover, SOSiA is reported to be non-hygroscopic compared to SOA (Janechek et al., 2019),
and we do not see an obvious relationship between the experiment humidity conditions and aerosol formation.





Figs. 4c and 4d show comparisons of the standard and aging-VBS with experimental SOSiA mass concentrations and $Y_{SOSiA}$.
We see an improvement in the $R^2$ with the aging-VBS over the standard-VBS model, suggesting that incorporating $OH_{exp}$ into
the yield parameterization improves model outcomes. Fig. 5 illustrates compares the standard-VBS model with the aging VBS
for a range of $OH_{exp}$, showing that product volatility gradually decreases with increasing $OH_{exp}$ in the ageing VBS model. The
high volatility of the initial products is consistent with the lack of the rapid formation of low-volatile species, like highly
oxygenated molecules, known to form SOA (Isaacman-VanWertz et al., 2018).

### 3.2.2 Reconciling Literature $Y_{SOSiA}$

To address the variation in the literature $Y_{SOSiA}$ and to generate parameters for air quality models, we fit the parameters in the
aging-VBS model with all available data in the literature and those from our experiments. Given that the literature used
differing $\rho_{SOSiA}$ to calculate $Y_{SOSiA}$ from SMPS data, we adjust the $Y_{SOSiA}$ and $C_{OA}$ reported in the literature to that of the $\rho_{SOSiA}$
used here ($\rho_{SOSiA} = 1.07$ g cm$^{-3}$). Similarly, we re-calculate the $OH_{exp}$ in the literature using Eq. (2) and the $[D_5]_0$ and $[D_5]_{final}$
values.

Fig. 6 shows experimental values (markers) and model outputs (contours) of $Y_{SOSiA}$ (panels a1 and a2) and SOSiA mass
concentrations (panels b1 and b2) as a function of $[D_5]_0$ and $OH_{exp}$. Figs. 6a1 and 6b1 are generated using the aging-VBS
model fit using only data from experiments presented in this study, while Figs. 6a2 and 6b2 show a fit including data from the
literature. The aging-VBS model is able to capture the increasing $Y_{SOSiA}$ with increasing $[D_5]_0$ and $OH_{exp}$. At a given $[D_5]_0$,
$Y_{SOSiA}$ and the SOSiA mass concentration increase with higher $OH_{exp}$. Fig. 6a2 shows that the relatively high $Y_{SOSiA}$ (> 50 %)
is feasible at $OH_{exp} > 10^{12}$ s cm$^3$. Moreover, the aging-VBS model predicts that $Y_{SOSiA}$ is almost negligible (< 5 %) under
atmospheric concentrations of $D_5$ and $OH_{exp}$.

Fig. S8 shows that the aging-VBS model used here leads to a much higher correlation between modelled and experimental
values for SOSiA mass concentration compared to the same analysis with a standard-VBS model ($R^2 = 0.956$ vs. $R^2 = 0.745$).
The better correlation suggests that the volatility distribution evolves with $OH_{exp}$ and that chemical aging should be considered
when evaluating the volatility distribution of SOSiA from $D_5$ + OH.

We note that, in reality, bulk-phase chemistry is more complex than logarithmic shifts in volatility with $OH_{exp}$ and not fully
captured in the above aging-VBS parameterization. For example, Wu and Johnston (2017), Avery et al. (2023), and Chen et
al. (2023) characterized $D_5$ + OH SOSiA with mass spectrometry and found spectra indicative of oligomers. The formation of
oligomers may reduce the bulk volatility by more than one bin and change the gas-particle equilibrium timescales (Berkemeier
et al., 2020). Here, we incorporated $k_{age,gas}$ and a simple "bin-hopping" approach to illustrate that a change in the volatility
distribution with $OH_{exp}$ can adequately reconcile the $Y_{SOSiA}$ variation in the literature. Future work with more sophisticated
chemical models should close that gap further.



**4 Conclusions and Atmospheric Implications**
With a PAM-OFR, PTR-MS, and SMPS, we studied the formation of VOP and SOSiA under various $OH_{exp}$ conditions. Using
a simplified VOP oxidation scheme (Eqs. (R5) and (R6)), we find that the VOP of tentatively identified siloxanols and formate
esters have shorter OH-oxidation lifetimes than their precursor $D_5$ (Table S7). In addition, we find the mass yield of HCHO of
$D_5$ comparable to that of isoprene or aromatics (Millet et al., 2006), suggesting that $D_5$ siloxane is a potential $O_3$-contributing
species in downwind scenarios. We find the mass yield of HCOOH ranging from 45 – 127 %, which suggests that $D_5$ + OH is
a source of atmospheric HCOOH.

An aging-VBS model incorporating $OH_{exp}$ and chemical aging adequately describes gas-particle partitioning at atmospheric
$OH_{exp}$ and $C_{OA}$. Based on these experiments, low-$NO_x$ $Y_{SOSiA}$ should be < 10 % under commonly observed atmospheric $OH_{exp}$
$< 5 \times 10^{11}$ s cm$^{-3}$ (Fig. 6a1). The first-generation products of $D_5$ + OH are likely volatile, but their volatility decreases with
increasing $OH_{exp}$ (Fig. 5). This shift in volatility suggests that further oxidation of secondary products would reduce the
volatility enough to form SOSiA. Unlike α-pinene (Isaacman-VanWertz et al., 2018) or other precursors for secondary organic
aerosol (SOA), $D_5$ + OH does not appear to produce low-volatile species within a single oxidation step. Instead, additional
$OH_{exp}$ is needed to form aerosol, which suggests that multiple oxidation steps lead to gradual decrease of product volatility.
Hence, concepts that can be successfully applied to SOA formation may not accurately capture SOSiA formation, for which
models must consider chemical aging. In the atmosphere, SOSiA from $D_5$ + OH may be easier to detect downwind of urban
sources due to the higher $OH_{exp}$ and dilution/removal of competing OH-reactive species.

Based on KinSim calculations (Sect. S5), we expect that the $RO_2$ fate is dominated by $RO_2$ + $HO_2$ and $RO_2$ + OH, which is
consistent with the calculations performed by Avery et al. (2023). However, we note that the reaction rate coefficients of $RO_2$
and its subsequent products are uncertain for $D_5$, and we cannot directly address the atmospheric relevance of these calculated
$RO_2$ fates at this time. To improve $Y_{SOSiA}$ parameterizations for the atmosphere, there is a need to study the impact $NO_x$ has on
siloxane $RO_2$ chemistry, given that siloxanes are likely emitted from urban sources where [$NO_x$] is high. In such scenarios,
$RO_2$ + $NO_x$ is likely an important fate (Peng et al., 2019; Newland et al., 2021). Han et al. (2022) found that the addition of
$N_2O$ into their OFR would reduce $Y_{SOSiA}$, although the cause is unclear. However, Charan et al. (2022) did not find $Y_{SOSiA}$ to
change with $NO_x$ in their chamber experiments, which is consistent with rapid RO formation across $RO_2$ fates. Quantifying
secondary species across $RO_2$ fates and identifying their subsequent oxidation reactions may also be useful to adapt the $D_5$
oxidation mechanism into chemical kinetics models.

The high [OH] used in OFRs may induce faster radical reactions and dimerization near the particle surface (Zhao et al., 2019),
which affects particle composition and equilibrium timescales. Dimers and oligomers have been found in SOSiA (Wu and
Johnston, 2017; Avery et al., 2023; Chen et al., 2023), and how oligomerization in the $D_5$ + OH SOSiA system evolves the



volatility distribution and particle properties is currently not considered in the aging-VBS model. Moreover, high degrees of
oxidation should lead to fragmentation and increasing volatility (Isaacman-VanWertz et al., 2018), which is also not considered
in the aging-VBS model. Hence, multiphase modeling to evaluate SOSiA chemistry and translate experimental findings to
atmospheric conditions remains a direction for future research.
**Appendix A Abbreviations**
$C_{OA}$: organic aerosol mass loading
C*: effective saturation mass concentration
$D_5$: decamethylcyclopentasiloxane
EVR: extended volatility range
ID: inner diameter of tubing
$I_{254}$, $I_{185}$: flux of 254 and 185 nm photons
OA: organic aerosol
OD: outer diameter of tubing
OFR: oxidation flow reactor
OH: hydroxyl radical
$[OH]_{avg}$: 24 hour average daily hydroxyl radical concentration
$OH_{exp}$: hydroxyl radical exposure
$OHR_{ext}$: external hydroxyl radical reactivity
$O_3$: ozone
ncps: normalized counts per second
$NO_x$: nitric oxide and nitrogen dioxide
PAM: potential aerosol mass
PTR: proton transfer reaction
PTR-MS: proton transfer reaction mass spectrometer
RH: relative humidity
RO: alkoxyl radical
$RO_2$: peroxyl radical
SMPS: scanning mobility particle sizer
SOA: secondary organic aerosol
SOSiA: secondary organosiloxane aerosol
UV: ultraviolet radiation
VBS: volatility basis set



VOP: volatile oxidation products
$Y_{HCHO}$: formaldehyde molar yield from $D_5$
$Y_{HCOOH}$: formic acid molar yield from $D_5$
$Y_{SOSiA}$: SOSiA mass yield from $D_5$
$\gamma$: molar yields extrapolated to when $OH_{exp} \rightarrow 0$
$\rho_{SOSiA}$: SOSiA aerosol mass density
$\tau_{res}$: residence time
**Data Availability**
Summary data are available in the supplementary. Additional data will be provided upon reasonable request.
**Author ORCID**
Hyun Gu Kang: https://orcid.org/0000-0002-3320-9447
Yanfang Chen: https://orcid.org/0000-0002-4415-7398
Jiwoo Jeong: https://orcid.org/0000-0002-4038-8148
Yoojin Park: https://orcid.org/0000-0002-4832-8633
Thomas Berkemeier: https://orcid.org/0000-0001-6390-6465
Hwajin Kim: https://orcid.org/0000-0001-6138-6443
**Author Contribution**
HGK, YC, JJ, and YP conducted the experiments. YP performed the offline calibrations of OH exposure on the PAM-OFR.
HGK analysed the data. HGK and TB developed the kinetic model. HGK, HK and TB wrote the manuscript with contributions
from all co-authors. HK supervised the project.
**Competing Interests**
TB is a member of the editorial board of Atmospheric Chemistry and Physics, but the peer-review process was guided by an
independent editor. The authors declare that they have no other personal nor financial conflicts of interest. Instruments and
products used in the research are listed for reference and not as endorsements.



## Acknowledgements

This work was supported by the FRIEND Project (Fine Particle Research Initiative in East Asia Considering National Differences), which is funded by the National Research Foundation of Korea (NRF) and the Ministry of Science and ICT of the Republic of Korea (2022M3G1A1020858). Also funded by National Research Foundation of Korea (NRF) funded by the Ministry of Science and ICT (NRF-2021R1A2C2004365). HGK is supported by the Max Planck Graduate Center with Johannes Gutenberg University Mainz. APM Engineering (Gyeonggi-do, South Korea) rent the PTR-MS, and the authors thank BO for maintaining it.

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





**Table 1. Summary of SOSiA mass yields ($Y_{SOSiA}$) with aerosol sampling line corrections assuming $\rho_{SOSiA}$ = 1.07 g cm⁻³ for all**
**experiments. [H₂O] is the molar mixing ratio of H₂O in air. For $C_{OA}$ and [D₅], the errors are the standard deviation of the data points**
**averaged, while for $Y_{SOSiA}$, they are calculated with error propagation. For reference, at 25 °C and 1 atm, 1 ppb of D₅ is ~15 µg m⁻³**
**and one day equivalent of $OH_{exp}$ is ~1.3 × 10¹¹ s cm⁻³ at a daily $[OH]_{avg}$ of 1.5 × 10⁶ cm⁻³.**

| Experiment | $Y_{SOSiA}$ (%) | [H₂O] (%) | $C_{OA}$ (µg m⁻³) | $OH_{exp}$ (s cm⁻³) | [OH] (cm⁻³) | $[D_5]_0$ (ppb) | $1 - [D_5]_{final}/[D_5]_0$ |
|---|---|---|---|---|---|---|---|
| 1 | $5.9 \pm 0.9$ | 0.892 | $10.5 \pm 0.7$ | $1.73 \times 10^{11}$ | $9.59 \times 10^{8}$ | $43.4 \pm 1.3$ | 0.292 |
| 2 | $4.9 \pm 0.6$ | 0.828 | $19.0 \pm 0.6$ | $1.90 \times 10^{11}$ | $1.06 \times 10^{9}$ | $85.7 \pm 2.5$ | 0.316 |
| 3 | $3.3 \pm 0.6$ | 0.742 | $17.7 \pm 0.5$ | $1.26 \times 10^{11}$ | $6.99 \times 10^{8}$ | $165.8 \pm 4.5$ | 0.222 |
| 4 | $19.5 \pm 1.5$ | 1.95 | $75.2 \pm 1.9$ | $4.66 \times 10^{11}$ | $2.59 \times 10^{9}$ | $44.0 \pm 1.7$ | 0.606 |
| 5 | $29.3 \pm 2.7$ | 2.06 | $179.2 \pm 3.1$ | $3.80 \times 10^{11}$ | $2.11 \times 10^{9}$ | $78.3 \pm 3.2$ | 0.532 |
| 6 | $26.5 \pm 1.8$ | 2.09 | $286.2 \pm 7.1$ | $3.12 \times 10^{11}$ | $1.73 \times 10^{9}$ | $157.8 \pm 3.6$ | 0.464 |
| 7 | $8.6 \pm 0.5$ | 0.733 | $36.8 \pm 1.3$ | $5.76 \times 10^{11}$ | $3.20 \times 10^{9}$ | $43.8 \pm 1.3$ | 0.684 |
| 8 | $18.6 \pm 1.7$ | 0.736 | $118.6 \pm 5.6$ | $4.00 \times 10^{11}$ | $2.22 \times 10^{9}$ | $78.9 \pm 3.2$ | 0.550 |
| 9 | $21.8 \pm 1.1$ | 0.797 | $304.5 \pm 2.8$ | $4.19 \times 10^{11}$ | $2.33 \times 10^{9}$ | $166.8 \pm 4.1$ | 0.567 |
| 10 | $39.8 \pm 2.2$ | 1.93 | $212.9 \pm 8.1$ | $9.01 \times 10^{11}$ | $5.00 \times 10^{9}$ | $43.8 \pm 1.4$ | 0.835 |
| 11 | $47.4 \pm 1.9$ | 2.08 | $420.2 \pm 3.0$ | $7.78 \times 10^{11}$ | $4.32 \times 10^{9}$ | $76.5 \pm 2.2$ | 0.789 |
| 12 | $54.0 \pm 2.4$ | 2.15 | $965.7 \pm 25$ | $7.39 \times 10^{11}$ | $4.10 \times 10^{9}$ | $156.9 \pm 3.9$ | 0.772 |
| 13 | $4.7 \pm 1.7$ | 0.712 | $3.9 \pm 0.3$ | $8.70 \times 10^{10}$ | $7.25 \times 10^{8}$ | $37.9 \pm 1.6$ | 0.160 |
| 14 | $1.9 \pm 0.4$ | 0.718 | $4.1 \pm 0.3$ | $1.09 \times 10^{11}$ | $9.10 \times 10^{8}$ | $80.8 \pm 2.3$ | 0.196 |
| 15 | $1.1 \pm 0.3$ | 0.704 | $3.7 \pm 0.7$ | $8.29 \times 10^{10}$ | $6.91 \times 10^{8}$ | $162.8 \pm 4.9$ | 0.153 |







**Figure 1. Example PTR-MS mass spectra from experiment 12 and proposed VOP ions.** The signal intensities, before (black) and after (grey) oxidation, are each normalized to the signal intensity of the $D_5$ ion at *m/z* 371, which is set to 1. The multifunctional species (blue, pink) are expected to be formed through multiple steps of OH-oxidation. The red and pink areas in the inset each refer to the enhancement in signal attributed to $D_4T$-OH and $D_3T_2$-$(OH)_2$ over that of the -$CH_3$ fragment of $D_5$ and isotope signals, respectively.

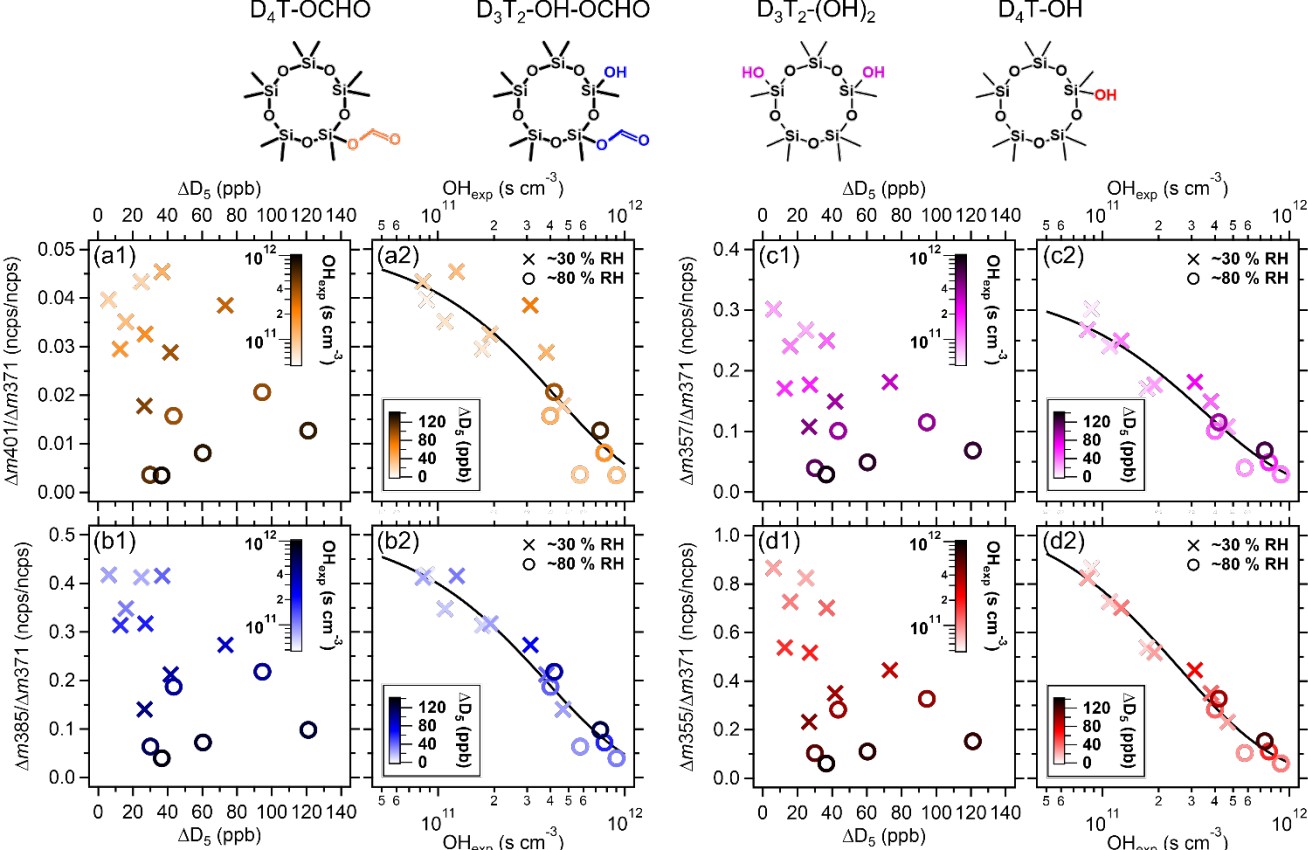

**Figure 2. Relative molar yields of selected VOP.** Molar yields as a function of $OH_{exp}$ and $D_5$ consumed in experiments for (a1, a2) $D_4T$-OCHO, (b1, b2) $D_3T_2$-OH-OCHO, (c1, c2) $D_3T_2$-$(OH)_2$, and (d1, d2) $D_4T$-OH. We did not have a calibration for the suspected VOP, so the y-axes are relative molar yields (ncps/ncps) calculated with the change in signal attributed to each VOP and that of $D_5$ at *m/z* 371. The relative molar yields decrease with $OH_{exp}$, which is used to fit their OH-oxidation rate coefficients and $\gamma_i$ (black lines).



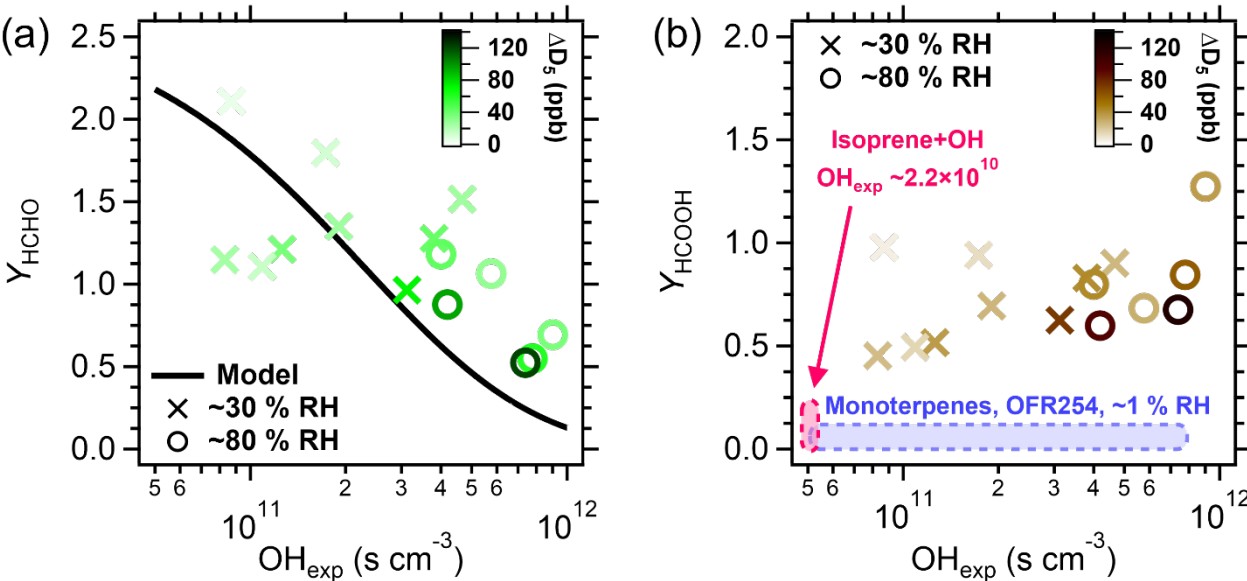

**Figure 3. Experimental molar yields of selected VOP: (a) HCHO and (b) HCOOH as functions of OH_exp.** The blue shaded area in (b) is the range of $Y_{HCOOH}$ (< 10 %) measured by Friedman and Farmer (2018) with monoterpenes under low RH and low $NO_x$ conditions. The pink shaded area refers to $Y_{HCOOH}$ from isoprene + OH chamber experiments (Link et al., 2020) at lower OH_exp.



**Figure 4. Application of standard-VBS and aging-VBS models to experimental data.** (a) $Y_{SOSiA}$ as a function of $C_{OA}$, where the $Y_{SOSiA}$ appears to be correlated with $OH_{exp}$. (b) VBS product mass yields for each volatility bin. For the aging-VBS, the values are those of the first-generation products. (c) Comparison of SOSiA mass concentrations from the aging-VBS and standard-VBS models against measurements. (d) Comparison of $Y_{SOSiA}$ from the aging-VBS and standard-VBS models against measurements, where the aging-VBS model has a higher $R^2$.












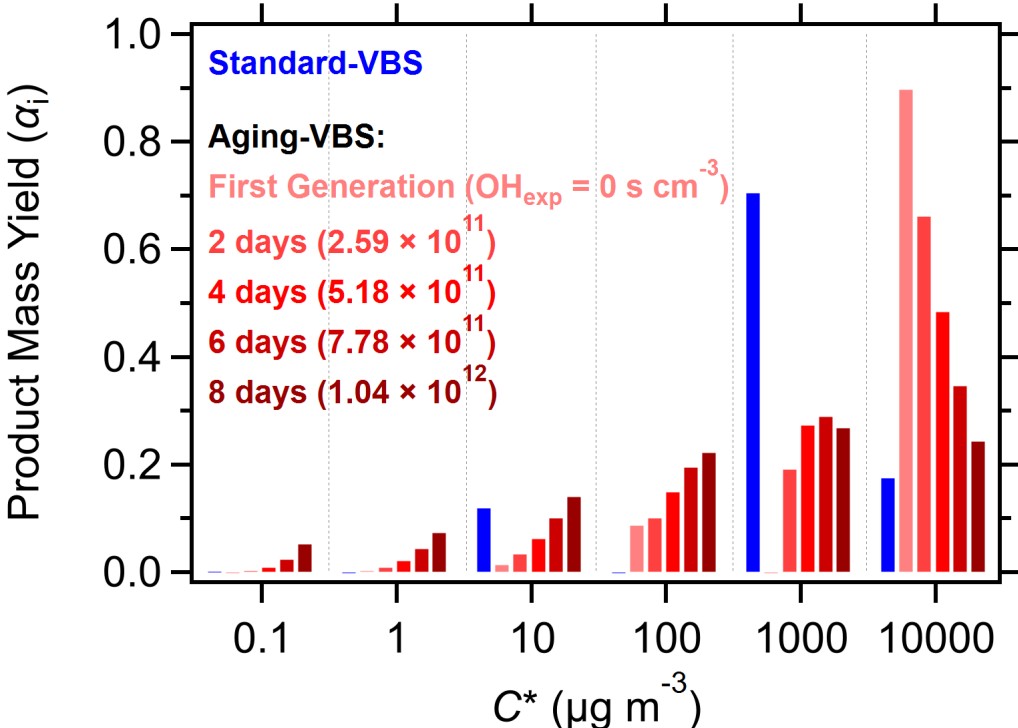

**Figure 5. Evolution of the volatility distribution with $OH_{exp}$.** The standard-VBS model parameterization (blue bars) is dominated by the $C^* = 1\,000$ μg m$^{-3}$ volatility bin. In the aging-VBS model, the first-generation volatility distribution is dominated by the highest volatility bin ($C^* = 10\,000$ μg m$^{-3}$) but decreases with increasing $OH_{exp}$ (red bars).

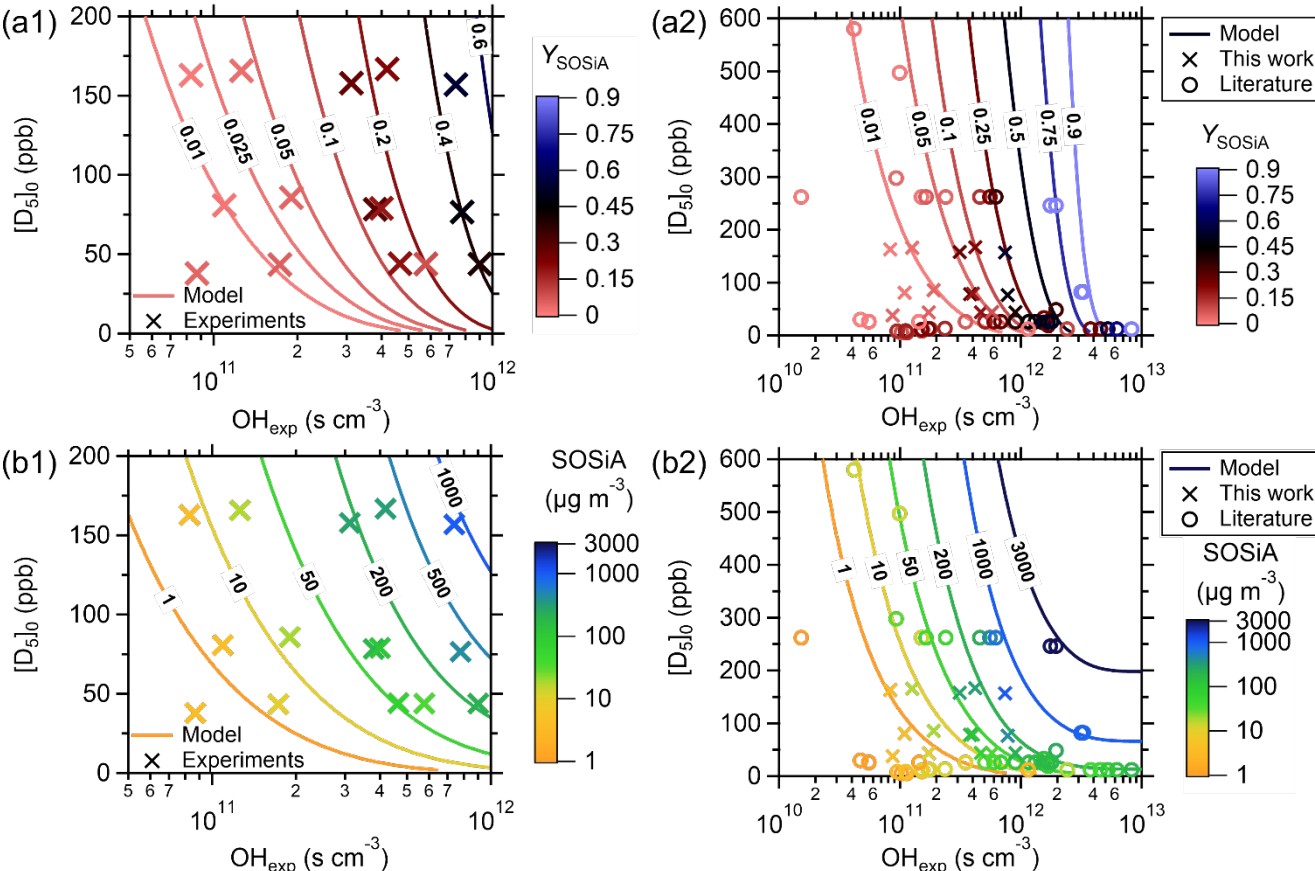

**Figure 6. Comparison of experiments, model results, and literature values.** (a) $Y_{SOSiA}$ and (b) SOSiA mass concentrations as a function of $[D_5]_0$ and $OH_{exp}$. The aging VBS-model is fit using experimental data from (1) this study and (2) including those in the literature. SOSiA formation generally increases with $[D_5]_0$ and $OH_{exp}$. The aging-VBS can capture the broad range of $Y_{SOSiA}$ reported in the literature.