# Peer review of "Volatile Oxidation Products and Secondary Organosiloxane Aerosol from $D_5$ + OH at Varying OH Exposures"

_EGUsphere, 2023_

## Author Comment (AC1)

We appreciate the anonymous reviewers for their thoughtful reviews and comments. We have carefully considered their suggestions and revised the manuscript accordingly. In addition to changes arising from the reviewers, we have made edits to correct figure references in the text and the figures themselves.

Key changes from the previous version are the separation of the gas and particle-phase chemical aging rate coefficients in the aging-VBS model and the addition of model sensitivity evaluations. With the revised model, we regenerate the model figures. We also include an ensemble of optimized parameter sets to provide a range of potential fit values and show the root mean square errors of the models against the data. Accordingly, the text has been updated so $k_{age,particle}$ is not tied to being 10 % of $k_{age,gas}$, including the abstract.

The reviewer comments are in blue, our comments are in black, and modifications to the manuscript are in red. Other edits and corrections not instigated by the reviewers are summarized at the end.

**Reviewer # 1**

General comments:

The authors investigated the photooxidation process of decamethylcyclopentasiloxane (D5), which is used in consumer products, using a flow reactor; the D5 oxidation products, i.e., silanols, formaldehyde, formic acid, and secondary aerosols, were measured revealing the photooxidation process in the gas and particle phase. By determining the variables of the volatility basis-set model, the measured yields of D5-derived secondary aerosols in this and previous studies were explained, and the mechanism by which the volatile silanols produced during the initial oxidation process undergo photochemical aging to form lower-volatility compounds which are partitioned into the particle phase was clarified. In particular the model-based explanation for D5-derived secondary aerosols is considered worthy of publication in the field of atmospheric chemistry. However, I hope the authors will read the following comments and consider revisions to the draft prior to publication.

We appreciate the anonymous reviewer for taking the time to thoroughly review our submission. We have summarized the revisions to improve the manuscript.

Specific comments:

Lines 106-110. Briefly describe the methods used in previous studies and discuss the implications of using the methods of this study to investigate the mechanism of photooxidation of D5, which has been interpreted differently in previous studies.

While the existing literature largely focused on the first-generation products from $D_5$ + OH (Fu et al., 2020), this manuscript expands that work by exploring the OH-oxidation rate coefficients of VOP via mass spectrometry and by quantifying HCHO, which is a suspected product. While Alton and Browne (2022) constructed a mechanism for the VOP, they did not have rate coefficients for their OH-oxidation. Consequently, we have edited this paragraph accordingly:

Revised lines 108-114:

"The intermediate products between $D_5$ and those small silanols are less studied, and the OH-oxidation rate coefficients of these intermediates have not been reported. Sommerlade et al. (1993) and Alton and Browne (2022) used mass spectrometry to study the gaseous products of $D_5$ oxidation in chambers, while Fu et al. (2020) used quantum chemistry modeling. These studies found that gaseous intermediates are composed of a variety of alcohols, aldehydes, esters, and hydroperoxides. Given that such volatile oxidation products (VOP) in experiments with higher $OH_{exp}$ are likely to undergo multiple oxidation steps, there is a need to address their subsequent oxidation rate coefficients. Moreover, while the formation of HCHO and HCOOH have been predicted in mechanisms, they have not been quantified."

Lines 127. There is only one sentence at the end of the introduction explaining the aim of the study. The topic of this study would not only be to parameterize aerosol yields. The initial product analysis of D5, the HCHO production yield measurement, and the formic acid production yield measurement to be performed in this study should also be reiterated, and the overall aims of this study should be summarized. The new measurements or calculations to be performed in this study should be again highlighted.

We thank the reviewer for this comment. To highlight the key aims of this paper, we have amended this sentence to a paragraph at the end of the introduction:

Revised lines 133-136:

"In this study, we aim to assess the OH-oxidation of $D_5$ by determining rate coefficients of secondary reactions of VOP with OH and providing a first quantification of HCHO and HCOOH yields. We also perform additional experiments to measure $Y_{SOSiA}$ under diverse $OH_{exp}$ and $[D_5]_0$. Lastly, we develop parameterizations using a kinetic model and a simplified chemical aging reaction scheme to reconcile the reported $Y_{SOSiA}$ from $D_5$ + OH in the literature for use in air quality models."

Line 150. By external OH reactivity, do you mean OH reactivity measured under the same conditions separately from the D5 reaction experiment? The meaning of the word "external" is ambiguous.

Thank you for this question. External OH reactivity ($OHR_{ext}$) refers to the OHR caused by species being injected into the PAM-OFR, which in these experiments would be from $D_5$. $OHR_{ext}$ is opposed to internal OHR ($OHR_{int}$), which is the OHR internal to the photochemistry of the PAM-OFR. This phrasing is used by Peng and Jimenez (2020), and we have opted to use it for consistency with the literature. To prevent confusion, we have modified the sentence to the following:

Revised lines 156-158:

"We performed the experiments with target $[D_5]_0$ of 50, 100, or 200 ppb. With these target $[D_5]_0$ we get external OH reactivities ($OHR_{ext}$) of $2.5 - 9.8$ $s^{-1}$ at 298.15 K and 1 atm, where $OHR_{ext}$ is the reactivity caused by the injection of $D_5$ into the PAM-OFR (Peng and Jimenez, 2020)."

Line 191. PerMaSCal ions are confusing; it would be easier to simply write diiodobenzene ions for the m/z 331 ions here.

We thank the reviewer for this comment and have removed the reference to PerMaSCal here to improve readability:

Revised lines 199-200:

> "We adjusted the MCP voltage in steps to increase the signal strength at *m/z* 331, a diiodobenzene ion, until the relative signal increase was < 20 %."

Lines 201-204 and section S2. Has a study been conducted to determine the effect of volatilization of particles from the filter? Also, you state that the accuracy of the balance is 0.1 mg, but if it is a semi-microbalance, wouldn't the accuracy be 0.01 mg? What was the mass of the particles collected by the filter?

Thank you for these comments. Since these filter collection experiments were conducted under humid conditions, we weighed the filters after placing them in a desiccator for 24 hours at room temperature to avoid weighing water. We do not have separate measurements to address volatiles evaporating off the collected particles, and this effect is an uncertainty in the measurement. We have added the following to Sect. S2:

Added supplementary lines 221-222:

> "Lastly, volatile species may have evaporated from the collected particles while the filters were in the desiccator, which would lead to an underestimation of the particle masses and thus SOSiA density."

Also, the reviewer is correct that the semi-microbalance accuracy should be ± 0.01 mg and the standard deviation of the density should accordingly be 0.08 g cm$^{-3}$. These mistakes have been fixed. We weighed each filter ten times before and after SOSiA collection and have measured particle masses in the range of 300 to 740 µg.

Revised supplementary lines 196-200:

> "Then, we stored the filter samples in a desiccator placed inside of a temperature and humidity-controlled micro-balance room for a day to remove mass interference from condensed water. Each filter was weighed ten times on a semi-micro balance (± 0.01 mg, ME204, Mettler Toledo, Columbus, OH, USA), and we calculate the mean $\rho_{SOSiA}$ by dividing the masses of SOSiA (300 – 740 µg) over integrated SMPS volumes."

Lines 246-247. Can we explain the experimental results of photochemical aging of D5-derived secondary aerosols without considering particle phase aging? There is no guarantee that the uptake factor will be the same as for the α-pinene SOA of Zhao et al. (2015); can we assess the sensitivity of a factor of 10% to the final fitting?

Thank you for this comment, which prompted us to do an in-depth sensitivity study of the photochemical aging rates. We find that the model is very sensitive to $k_{age,particle}$, with a higher rate coefficient resulting in higher SOSiA production (Fig. S11a). We also performed a global optimization without particle-phase aging (green markers in Fig. S11b) causing the model to perform worse (RMSE = 55.8) and the fitted product volatility distribution to differ (Fig. S11c). Hence, we performed another global optimization using a free fit in both aging rates (purple markers in Fig. S11b). We find a very similar $k_{age,particle}$ of ~$2\times10^{-12}$ cm$^{-3}$ s$^{-1}$, but a slower $k_{age,gas}$ of ~$1\times10^{-12}$ cm$^{-3}$ s$^{-1}$, leading to a slight improvement in model performance (RMSE = 42.6) compared to the original fit (RMSE = 44.2).

Both fitted reaction rate coefficients are plausible as $k_{age,gas}$ is now comparable to the reaction rate coefficient of D$_5$ and OH, and $k_{age,particle}$ is still below the collision limit of OH with the particle surface (i.e. a hypothetical case of an uptake coefficient of 1). To determine the latter, we calculate the collision flow ($F_{coll,OH}$) of OH radicals with the aerosol surface for each experiment's final aerosol size distribution. In the equation below, $\omega_{OH}$ is the mean thermal velocity of OH and $A_{particle}$ is the particle surface area density.

$$F_{coll,OH} = \frac{\omega_{OH}}{4} A_{particle} [OH]$$

The heterogeneous reaction flow of OH with the particle surface ($F_{het,OH}$) cannot be larger than $F_{coll,OH}$ as particles will not uptake all OH colliding their surfaces. In the below equation for $F_{het,OH}$, $c_{particle}$ is the concentration of total condensed species.

$$F_{het,OH} = k_{age,particle} \, c_{particle} \, [OH]$$

By rearranging the equations for $F_{coll,OH}$ and $F_{het,OH}$, we arrive at an expression for the upper limit of $k_{age,particle}$:

$$k_{age,particle} < \frac{\omega_{OH}}{4 \, c_{particle}} A_{particle}$$

With this method, we find a physical maximum for $k_{age,particle}$ of $2 - 7 \times 10^{-12}$ cm$^{-3}$ s$^{-1}$, depending on the particles' surface area density (Table S12). Hence, with $k_{age,particle} = 2 \times 10^{-12}$ cm$^{-3}$ s$^{-1}$, the effective uptake coefficient of OH on SOSiA is less than one.

We also separately analyze a fit ensemble of parameter sets where the model RMSE is less than 50 generated from Monte Carlo sampling during the global optimization, a method described in

Berkemeier et al. (2021).We find that the volatility distribution of the fit ensemble is consistent with the best fit, with most of the product mass yield in the highest volatility bins (Fig. 12a).

The aging-VBS model is less sensitive to $k_{age,gas}$, as indicated by a large variability in the fit ensemble, while the numerical value of $k_{age,particle}$ is narrowly constraint (Fig. S12b). We now use the updated photochemical aging rate coefficients for all calculations in the paper and added the following discussions to the manuscript:

Added lines 260-264:

"We fit $k_{age,gas}$ , $k_{age,particle}$, and $\alpha_i$ in the aging-VBS model to the experimental SOSiA mass using the Monte Carlo genetic algorithm (MCGA) (Berkemeier et al., 2017). We obtain a best model fit and a fit ensemble consisting of 548 parameter sets for which the model's root mean square error (RMSE) is below a threshold of 50. We find this ensemble to estimate the parametric uncertainty associated with the model fit (Berkemeier et al., 2021)."

Added lines 427-433:

"We find that the model is very sensitive to $k_{age,particle}$, as a higher $k_{age,particle}$ will result in higher model SOSiA formation (Fig. S11a), but not sensitive to $k_{age,gas}$. In addition, $k_{age,particle}$ is tightly constraint in the ensemble of model fits around a value of $2 \times 10^{-12}$ cm$^3$ s$^{-1}$ (Fig. S12b). When fitting the model with deactivated particle-phase aging ($k_{age,particle}$ = 0), model-experiment RMSE is significantly increased and the fitted $k_{age,gas}$ becomes unphysically large. The numerical value of the fitted $k_{age,particle}$, on the other hand, is physically reasonable as it corresponds to an effective uptake coefficient of OH molecules colliding with the particle surface of less than one (Sect. S1.6). We hence postulate that multi-generational aging of SOSiA occurs predominantly in the particle phase."

Added lines 492-493:

"We also find that the aging-VBS model is sensitive to $k_{age,particle}$ (Fig. S11) and not sensitive to $k_{age,gas}$ (Fig. S12), suggesting that heterogeneous aging should be considered in these models."

Added supplementary Sect. S1.6 Upper Limit Estimation of $k_{age,particle}$:

"To address whether the numerical value of the fitted $k_{age,particle}$ is reasonable, we calculate its upper physical limit as the collision flow of OH onto the particles in one cm$^3$ of air ($F_{coll,OH}$, cm$^{-3}$ s$^{-1}$) derived from gas kinetic theory (Pöschl et al., 2007).

$$F_{coll,OH} = \frac{\omega_{OH}}{4} A_{particle}[OH] \tag{S8}$$

Here, $\omega_{OH}$ is the mean thermal velocity of OH in cm s$^{-1}$ (Eq. (S6)) and $A_{particle}$ is the particle surface area density (cm$^2$ cm$^{-3}$) measured with the SMPS at the outlet of the PAM-OFR. This flow must always be larger than the heterogeneous reaction flow of OH with the particle surface in one cm$^3$ of air ($F_{het,OH}$, cm$^{-3}$ s$^{-1}$).

$$F_{het,OH} = k_{age,particle} \, c_{particle} \, [OH] \tag{S9}$$

Accordingly, we find the following condition for $k_{age,particle}$.

$$k_{age,particle} < \frac{\omega_{OH}}{4\,c_{particle}} A_{particle} \qquad\qquad (S10)$$

Here, $c_{particle}$, (cm$^{-3}$) denotes the concentration of total SOSiA products in the particle phase in one cm$^3$ of air. The estimated upper limit $k_{age,particle}$ are summarized in Table S12."

Similarly, the panels in Fig. 2 have been updated, and the main text reflects those changes.

Revised lines 398-403:

"We fit $k_{age,gas}$ and $k_{age,particle}$ in the aging-VBS model to be 1.14 × 10$^{-12}$ cm$^3$ s$^{-1}$ and 2.18 × 10$^{-12}$ cm$^3$ s$^{-1}$ respectively. The fitted aging-VBS model parameters are summarized in Table S11. Fig. 4a also shows the aerosol mass yield curves calculated with the aging-VBS model over varying OH$_{exp}$. Since the aging-VBS model is kinetic, the $Y_{SOSiA}$ are dependent on both [D$_5$]$_0$ and OH$_{exp}$, and we calculate three yield curves using the approximate experimental [D$_5$]$_0$. The yield curves generated with the aging-VBS model are more consistent with the experiments and show how $Y_{SOSiA}$, [D$_5$]$_0$, and OH$_{exp}$ are intertwined in the proposed aging mechanism."

Added lines 414-417:

"The optimized $\alpha_i$ for the aging-VBS model are shown as markers in Fig. 4b. The error bars indicate the minimum and maximum values of the fitted $\alpha_i$ in the ensemble parameter sets, which are further expanded in Fig. S12a. The fit ensemble suggest that products from D$_5$ + OH must be largely volatile (C* ≥ 10$^3$ μg m$^{-3}$) in order to reproduce the experimental SOSiA yields."

Revised lines 419-422:

"Figs. 4c and 4d show comparisons of the standard and aging-VBS models with experimental SOSiA mass concentrations and $Y_{SOSiA}$. The error bars indicate the range of model outcomes within the fit ensemble. We see an improvement in the RMSE and R$^2$ with the aging-VBS over the standard-VBS model, suggesting that incorporating OH$_{exp}$ into the yield parameterization improves model performance."

Added supplementary figures and table:

[Figure]

Figure S11. Sensitivity of the aging-VBS model to $k_{age,particle}$: (a) SOSiA formation at varying $k_{age,particle}$ assumptions compared using the optimized parameters found with the "base" model. The base model refers to the assumption where $k_{age,particle}$ = 0.1 × $k_{age,gas}$, which is sometimes used in the literature (Zhao et al., 2015). (b) SOSiA formation using optimized parameters found under each $k_{age,particle}$ assumption. The optimized parameters produce similar RMSE for each corresponding $k_{age,particle}$ assumption. (c) VBS found under each $k_{age,particle}$ assumption. The product mass yields vary only slightly except in the case where $k_{age,particle}$ = 0, while $k_{age,gas}$ changes. The purple markers and bars in panels (b) and (c) are from the aging-VBS model used in this paper where $k_{age,gas}$ and $k_{age,particle}$ are fit separately.

[Figure]

Figure S12. Range of the optimized parameter sets of (a) $\alpha_i$ and (b) the chemical aging rate coefficients in the fit ensemble. "Optimized" values refer to the best-fit parameter set found with MCGA global optimization. During the global optimization, we generate 768 000 Monte Carlo samples with randomly assigned parameter values. Then, we find the parameter sets where the aging-VBS model outputs RMSE < 50 against SOSiA measurements. We find small variation in the $k_{age,particle}$ while there is large variation in $k_{age,gas}$, suggesting the model is less sensitive to $k_{age,gas}$.

Table S12. Experiment particle surface area densities and OH to particle collision flow, which we denote as the maximum potential $k_{age,particle}$. These maximum values are calculated assuming unity uptake of OH, and the surface area densities are calculated using the particle size distributions measured with the SMPS.

| Experiment | Surface Area Density (cm² cm⁻³) | Max $k_{age,particle}$ (cm³ s⁻¹) |
|---|---|---|
| 1 | $6.23 \times 10^{-6}$ | $5.50 \times 10^{-12}$ |
| 2 | $9.84 \times 10^{-6}$ | $4.78 \times 10^{-12}$ |
| 3 | $9.56 \times 10^{-6}$ | $4.98 \times 10^{-12}$ |
| 4 | $3.48 \times 10^{-5}$ | $4.25 \times 10^{-12}$ |
| 5 | $6.41 \times 10^{-5}$ | $3.30 \times 10^{-12}$ |
| 6 | $8.64 \times 10^{-5}$ | $2.79 \times 10^{-12}$ |
| 7 | $1.69 \times 10^{-5}$ | $4.23 \times 10^{-12}$ |
| 8 | $3.71 \times 10^{-5}$ | $2.89 \times 10^{-12}$ |
| 9 | $7.22 \times 10^{-5}$ | $2.19 \times 10^{-12}$ |
| 10 | $7.63 \times 10^{-5}$ | $3.31 \times 10^{-12}$ |
| 11 | $1.24 \times 10^{-4}$ | $2.73 \times 10^{-12}$ |
| 12 | $2.28 \times 10^{-4}$ | $2.19 \times 10^{-12}$ |
| 13 | $2.97 \times 10^{-6}$ | $7.00 \times 10^{-12}$ |
| 14 | $2.95 \times 10^{-6}$ | $6.57 \times 10^{-12}$ |
| 15 | $2.42 \times 10^{-6}$ | $6.02 \times 10^{-12}$ |

Lines 254-256. Why normalize the mass spectra before and after the reaction by the signal of m/z 371, which is the signal of D5? The signal of m/z 371 after OH exposure should be decreased by the reaction

than before the exposure. How much did m/z 371 decrease by OH exposure in Experiment 12? It seems meaningless to compare before and after reaction by normalizing by m/z 371 signal without considering the decrease of m/z 371 signal intensity by reaction.

The y-axis of the mass spectra in Fig. 1 is scaled relative to the signal of m/z 371 to show how the mass spectra changed before and after oxidation. This scaling is done only for this visualization and not used in the quantification of $D_5$. We have edited this paragraph and rearranged the text to prevent confusion:

Revised lines 273-289:

"In Fig. 1, the PTR-MS signals before and after $D_5$ is oxidized are displayed relative to the protonated $D_5$ ion at *m/z* 371 on the y-axis. We perform this scaling because the isotopologues of the product fragment ions overlap with the isotopologues of $D_5$. Thus, changes in signal intensity are caused by both product formation and $D_5$ oxidation. We choose to normalize the spectra at *m/z* 371 because we assume that no product ion peaks overlap with the $[D_5]H^+$ signal at *m/z* 371. While this scaling makes the product peaks appear larger, the changes in the mass spectrum are also qualitatively highlighted. For example, $D_5$ loses a methyl group during the PTR which forms a large signal at *m/z* 355. The isotopologues of the -$CH_4$ fragment of $[D_5]H^+$ overlap with fragments of VOP. By scaling the mass spectrum with the ratio of $[D_5]H^+$ signal before and after oxidation, the signal of the VOP is separated from that of remaining $D_5$.

Using the mass spectra and species reported by Alton and Browne (2022), we attribute the indicated ions in Fig. 1 to siloxanol ($D_4T(OH)$), siloxanediol ($D_3T_2(OH)_2$), siloxanyl formate ($D_4T(OCHO)$), and siloxanolyl formate ($D_3T_2(OH)(OCHO)$). Here, "D" and "T" refers to silicon centers bonded to two and three oxygen atoms respectively. The multifunctional VOP are reported to arise from multiple steps of oxidation (Alton and Browne, 2022). The red and pink shaded areas in the inset of Fig. 1 refer to the enhancement in signal over that of the -$CH_4$ fragment of $[D_5]H^+$, which we attribute to the -$H_2O$ fragments of $[D_4T(OH)]H^+$ and $[D_3T_2(OH)_2]H^+$, respectively. We use the masses of the -$H_2O$ fragments of the protonated siloxanols as large alcohols dissociate during the PTR (Brown et al., 2010). We also attribute the ions in the blue and yellow-dotted boxes to the -$H_2O$ fragments of $[D_3T_2(OH)(OCHO)]H^+$ and $[D_4T(OCHO)]H^+$."

Lines 261-266. Since the m/z ratio of protonated D5 (D5-H+) produced in the PTR is m/z 371, the formation of the ion at m/z 355 should be described as loss of methane from protonated D5 ([D5-H - CH4]+) rather than loss of methyl from D5. Similarly, the loss of OH from silanol should be expressed as loss of water molecules from protonated silanol.

We thank the reviewer for this suggestion. To reflect that the fragmentation is of the protonated molecule, we have now corrected the text in the style as shown below and applied similar edits to the supplementary. We denote ions in square brackets and functional groups are in parentheses.

Revised lines 285-289:

"The red and pink shaded areas in the inset of Fig. 1 refer to the enhancement in signal over that of the -$CH_4$ fragment of $[D_5]H^+$, which we attribute to the -$H_2O$ fragments of $[D_4T(OH)]H^+$ and $[D_3T_2(OH)_2]H^+$, respectively. We use the masses of the -$H_2O$ fragments of the protonated

siloxanols as large alcohols dissociate during the PTR (Brown et al., 2010). We also attribute the ions in the blue and yellow-dotted boxes to the -$H_2O$ fragments of $[D_3T_2(OH)(OCHO)]H^+$ and $[D_4T(OCHO)]H^+$."

Lines 276-278. "Consistent" may be an error for "constant". In the same sentence, if the absorption of VOPi into the particle phase is discussed and then ignored, evidence should be provided that it can be ignored.

Thank you very much and yes, "consistent" should be "constant", and we have corrected this error. We suspect that the first-generation VOP will not condense into particles based on the calculations reported by Alton and Browne (2022), who used a quantitative structure activity relationship model to estimate their volatilities. We have added this citation in this sentence for clarity.

Revised lines 300-302:

"Then, assuming [OH] is constant throughout the PAM-OFR, that $D_5$ + OH is the rate-limiting step in VOP formation, and that removal via gas-particle partitioning is negligible (Alton and Browne, 2022), we can consider a simplified $D_5$ + OH chemical mechanism, Eqs. (R4) and (R5)."

Line 304. The abbreviation ODE is only used in one place in the text. It would be easier to understand if the abbreviation were not used and the term "ordinary differential equations" were used again here.

We have removed references to "ODE" throughout the manuscript to improve readability. The original line the reviewer refers to here has been removed during the revision to address the comment below.

Lines 307-310. The formation of formaldehyde by subsequent oxidation is ignored in Equations 7 and 8, even though it is later considered that formaldehyde is formed by subsequent oxidation. In fact, all experimental results for OHexp=3E11-1E12 in Figure 3 are higher than the value of the fitted curve. This is probably due to the failure to account for the subsequent formation of HCHO. Fitting with an incorrect model could provide data with systematic errors for the determined γHCHO.

We agree with the reviewer that the consecutive oxidation of the VOP should be considered to accurately model HCHO formation. Consequently, we implement a mechanism where the formation of HCHO occurs over multiple reaction steps. However, how VOP + OH branches to produce HCHO and the rate coefficients for those reactions is not well constrained. Thus, we opt to use a simplified mechanism where $D_5$ + OH produces a representative VOP ($VOP_{rep}$) and yields HCHO at each oxidation step. The subsequent VOP + OH reactions share the same rate coefficient as $D_5$ + OH and produces HCHO with the same yield ($\gamma_{HCHO}$).

$$D_5 + OH \rightarrow VOP_{rep} + \gamma_{HCHO}\ HCHO$$

$$VOP + OH \rightarrow VOP_{rep} + \gamma_{HCHO}\ HCHO$$

$$HCHO + OH \rightarrow$$

With this multi-step oxidation scheme, we find that the model better fits the data (red line) and a $\gamma_{HCHO}$ of 2.23. We have updated the manuscript accordingly.

Revised lines 327-342:

"However, HCHO formation likely occurs over multiple oxidation steps (Fu et al., 2020), and how VOP + OH branches to produce HCHO and the rate coefficients for those reactions is not experimentally constrained.

Consequently, we implement a simplified mechanism (Eqs. (R6) – (R8)), where $D_5$ + OH produces a representative VOP ($VOP_{rep}$) and yields HCHO at each oxidation step. The subsequent $VOP_{rep}$ + OH reactions share the same rate coefficient as $D_5$ + OH and produces HCHO with the same yield ($\gamma_{HCHO}$). This $\gamma_{HCHO}$ is the cumulative molar yield of HCHO, or the molar yield of HCHO as $OH_{exp} \rightarrow$ 0. This $\gamma_{HCHO}$ is also used to correlate satellite column retrievals of HCHO with VOC emissions (Millet et al., 2006), where an empirical value can be used to constrain uncertainty.

$$D_5 + OH \rightarrow VOP_{rep} + \gamma_{HCHO}HCHO \qquad (k_{D5+OH} = 2.0 \times 10^{-12} \text{ cm}^3 \text{ s}^{-1}) \qquad (R6)$$

$$VOP_{rep} + OH \rightarrow VOP_{rep} + \gamma_{HCHO}HCHO \qquad (k_{D5+OH}) \qquad (R7)$$

$$HCHO + OH \rightarrow \qquad (k_{HCHO+OH} = 8.5 \times 10^{-12} \text{ cm}^3 \text{ s}^{-1}) \qquad (R8)$$

We fit $\gamma_{HCHO}$ to be 223 % (black line in Fig. 3a), assuming a constant [OH] in the PAM-OFR and that HCHO removal via partitioning or reactive uptake is negligible."

[Figure]

Revised Figure 3:

[Figure]

"Figure 3. Experimental molar yields of selected VOP: (a) HCHO and (b) HCOOH as functions of $OH_{exp}$. The blue shaded area in (b) is the range of $Y_{HCOOH}$ (< 10 %) measured by Friedman and Farmer (2018) with monoterpenes under low RH and low $NO_x$ conditions. The pink shaded area refers to $Y_{HCOOH}$ from isoprene + OH chamber experiments (Link et al., 2020) at lower $OH_{exp}$."

Lines 344-352. This paragraph ultimately only compares the measured formic acid yield from D5 to that from isoprene. what evidence is there to conclude that D5 should be considered as a source of formic acid in the atmosphere? The current explanation is inadequate.

The reviewer is correct, global isoprene emissions are estimated to be ~500 Tg $yr^{-1}$ (Guenther et al., 2012), while that of $D_5$ is 0.03 Tg $yr^{-1}$ (McLachlan et al., 2010). Based on these estimates, the global HCOOH contribution from $D_5$ siloxane is likely small compared to isoprene, even if the yield is higher. Consequently, we have modified the text to qualify how much $D_5$ may contribute to global HCOOH:

Revised lines 385-388:

> " While $D_5$ + OH may produce more HCOOH than isoprene + OH, the global emissions of $D_5$ (McLachlan et al., 2010) are about four orders of magnitude smaller than those of isoprene (Guenther et al., 2012). Nevertheless, the product class of siloxanes may constitute a minor atmospheric HCOOH source in urban locations, especially if emissions were to increase."

Line 424. KinSim, which is not mentioned in the text, is suddenly mentioned in the conclusion. Section S5 and KinSim should be briefly explained in advance at appropriate places in the text.

Thank you for this comment. We use KinSim to address the possibility of alternate $RO_2$ pathways, explaining the variation in $Y_{SOSiA}$, but the model suggests that $RO_2$ fate was uniform across these experiments. We have added the following to Sect. 2.3:

Added lines 266-268:

> "We use the OFR chemistry template with KinSim (Peng and Jimenez, 2020) to estimate the $RO_2$ fates and expect the fates to have been uniform across the experiments (Sect. S5). Although

there are uncertainties in the RO₂ reaction rate coefficients for siloxanes, we expect that the variation in $Y_{SOSiA}$ is not driven by RO₂ fate in these experiments."

Fig. 1. The figure contains the structural formula of the analyte. However, since the figure is a mass spectrum, it may be necessary to provide an explanation of the detected ions instead of an explanation of the analyte. For example, you could simply indicate the symbols A-E in the figure and add the following explanation to the figure title; A: [D5-H]+, B: [D3T2-OH-OCHO-H - CH4]+, C: [D4T-OCHO-H]+, D: [D4T-OH-H - H2O]+, and E: [D43T2-(OH )2-H - H2O]+.

We have updated the figure to indicate the specific ions to which we attribute the VOP. In addition, we have adjusted the labeling on the y-axes to $Y_{rel,VOP}$ to clarify that these are the relative molar yields. Similar adjustments have been made to the text.

Modified Fig. 2 and caption:

[Figure]

"Figure 2. Relative molar yields of VOP as a function of $OH_{exp}$ and $D_5$ consumed. (a1, a2) $D_4T(OCHO)$, (b1, b2) $D_3T_2(OH)(OCHO)$, (c1, c2) $D_3T_2(OH)_2$, and (d1, d2) $D_4T(OH)$. The colors correspond to the attributed mass ions and molecular structures shown at the top. We did not have a calibration for the suspected VOP, so the y-axes are relative molar yields (ncps/ncps) calculated with the change in signal attributed to each VOP and that of $D_5$ at $m/z$ 371. The relative molar yields decrease with $OH_{exp}$, which is used to fit their OH-oxidation rate coefficients and $\gamma_i$ (black lines)."

Fig. 2. Explain in words somewhere that the color used for the functional group of the compound corresponds to the color of the plot.

We have used the colors to highlight the functional group changes on the $D_5$ backbone and to attribute the mass spectra signal to each VOP in Fig 1. We have amended the caption for Fig. 2, as shown above, and the below text to better explain the colors in Figs. 1 and 2:

Revised lines 285-289:

> "The red and pink shaded areas in the inset of Fig. 1 refer to the enhancement in signal over that of the -CH$_4$ fragment of [D5]H$^+$, which we attribute to the -H$_2$O fragments of [D$_4$T(OH)]H$^+$ and [D$_3$T$_2$(OH)$_2$]H$^+$, respectively. We use the masses of the -H$_2$O fragments of the protonated siloxanols as large alcohols dissociate during the PTR (Brown et al., 2010). We also attribute the ions in the blue and yellow-dotted boxes to the -H$_2$O fragment of [D$_3$T$_2$(OH)(OCHO)]H$^+$ and [D$_4$T(OCHO)]H$^+$."

Fig. 4b. Explain in words somewhere that the light colored area represents αi for gas and the dark colored area represents αi for particles. Does this figure show the gas-particle distribution ratio for COA=10ug/m3? If so, it should be half gas and half particles in 10^1ug/m3 bins. Correct any errors in calculations or markings.

We agree with the reviewer that the log scaling on the y-axis is confusing; because the figure is shown in a log scale on the y-axis, that half of the product in the $C^*$ = 10 μg m$^{-3}$ bin is condensed is not easily seen. We have revised the panel to be on a linear scale. Given that most of the products would be in the gas phase anyways when $C_{OA}$ = 10 μg m$^{-3}$, we have removed the dark/light shading for clarity. Furthermore, we have updated Fig. 4 with the new model results, including an error estimate using the fit ensemble.

Revised Fig. 4:

[Figure]

"Figure 4. Application of standard-VBS and aging-VBS models to experimental data. (a) $Y_{SOSiA}$ as a function of $C_{OA}$, where the $Y_{SOSiA}$ appears to be correlated with $OH_{exp}$. The standard-VBS model is shown in blue, and the aging-VBS model is shown with $OH_{exp}$ (color scale) as it is a kinetic model. (b) VBS product mass yields for each volatility bin. For the aging-VBS, the values are those of the first-generation products. (c) Comparison of SOSiA mass concentrations and (d) comparison of $Y_{SOSiA}$ between the aging-VBS and standard-VBS models against measurements. The error bars indicate the minimum and maximum values from the parameter fit ensemble. The aging-VBS model shows a lower RMSE and higher $R^2$."

**Additional Manuscript Changes**

**Graphical Abstract:** We have replaced the original figure to the one below, which removes the reference to $RO_2$, whose pathways are not considered in the proposed aging-VBS model.

[Figure]

**Correction to "D" and "T" in the molecular formulas.** Previously, we stated that "D" and "T" refer to units of $(CH_3)_2SiO$ and $CH_3SiO$ respectively. We have corrected explanations of this nomenclature throughout the manuscript for consistency with the literature: "D" and "T" refer to silicon center atoms being bonded with two and three oxygens respectively.

**Changed the title of Sect. 3.2.2 from "Reconciling Literature $Y_{SOSiA}$" to "Consolidating Literature $Y_{SOSiA}$" to prevent misinterpretation.** While the reported SOSiA mass yields vary between papers, they can be consolidated with a single aging-VBS model.

**Added model explanation on how experiment temperature variation was accounted for.** We add the below model detail for replicability.

Added lines 226-228:

> "Since the experiments had slight variations in temperature, we correct for temperature impacts on $C*$ between experiments using the Clausius-Clapeyron equation and an enthalpy of vaporization of 60 kJ mol$^{-1}$, which is that of $D_5$ siloxane (Lei et al., 2010)."

**Corrected condensational sink values in Table S3:** We previously misstated the $CS$ to be in units of m$^{-2}$, while it should be in units of s$^{-1}$ and miscalculated the $\tau_{CS}$ to be too fast. Eqs. S1 and S2 have been corrected, and the values in Table S3 have been revised.

Revised Table S3:

> "Table S3. Summary of experiment condensational sinks, LVOC condensation lifetimes, and growth factors calculated with the particle size distribution exiting the PAM-OFR as described in Section S1.3."

| ameters | SOA, LVOC $\kappa = 0.13$, $M = 0.200$ kg mol$^{-1}$ | | | SOSiA, $D_5$ $\kappa = 0.01$, $M = 0.370$ kg mol$^{-1}$ | | |
|---|---|---|---|---|---|---|
| eriment | $CS$ (s$^{-1}$) | $\tau_{CS}$ (s) | Growth Factor | $CS$ (s$^{-1}$) | $\tau_{CS}$ (s) | Growth Fac |
| 1 | $2.57 \times 10^{-2}$ | 38.8 | 1.02 | $1.88 \times 10^{-2}$ | 53.3 | 1.00 |
| 2 | $3.99 \times 10^{-2}$ | 25.1 | 1.02 | $2.92 \times 10^{-2}$ | 34.3 | 1.00 |
| 3 | $3.88 \times 10^{-2}$ | 25.76 | 1.02 | $2.84 \times 10^{-2}$ | 35.2 | 1.00 |
| 4 | 0.173 | 5.77 | 1.17 | 0.101 | 9.85 | 1.02 |
| 5 | 0.303 | 3.30 | 1.17 | 0.182 | 5.50 | 1.02 |

| | | | | | | |
|---|---|---|---|---|---|---|
| 6 | 0.394 | 2.54 | 1.17 | 0.239 | 4.19 | 1.02 |
| 7 | $6.68 \times 10^{-2}$ | 15.0 | 1.02 | $4.95 \times 10^{-2}$ | 20.2 | 1.00 |
| 8 | 0.138 | 7.27 | 1.02 | 0.104 | 9.63 | 1.00 |
| 9 | 0.250 | 3.99 | 1.02 | 0.192 | 5.20 | 1.00 |
| 10 | 0.338 | 2.95 | 1.12 | 0.217 | 4.61 | 1.01 |
| 11 | 0.522 | 1.92 | 1.12 | 0.342 | 2.93 | 1.01 |
| 12 | 0.913 | 1.09 | 1.12 | 0.605 | 1.65 | 1.01 |
| 13 | $1.25 \times 10^{-2}$ | 80.1 | 1.02 | $9.06 \times 10^{-3}$ | 110 | 1.00 |
| 14 | $1.23 \times 10^{-2}$ | 81.3 | 1.02 | $8.96 \times 10^{-3}$ | 112 | 1.00 |
| 15 | $1.01 \times 10^{-2}$ | 99.3 | 1.02 | $7.35 \times 10^{-3}$ | 136 | 1.00 |

**Corrected $Y_{HCHO}$ error values in Table S8.** An error in the error propagation calculation resulted in the errors being overstated.

Revised Table S8"

"Table S8. Experimental molar yields of HCHO and HCOOH. As these species are formed in the OFR at an unknown point, there may be some loss through oxidation with OH. Consequently, the OHexp determined with $D_5$ may not represent the $OH_{exp}$ these VOP experienced."

| Experiment | ΔHCHO/ΔD$_5$ (ppb/ppb) | ΔHCOOH/ΔD$_5$ (ppb/ppb) |
|---|---|---|
| 1 | 1.79 ± 0.25 | 0.94 ± 0.15 |
| 2 | 1.35 ± 0.15 | 0.69 ± 0.09 |
| 3 | 1.21 ± 0.21 | 0.52 ± 0.09 |
| 4 | 1.52 ± 0.11 | 0.90 ± 0.09 |
| 5 | 1.28 ± 0.11 | 0.83 ± 0.09 |
| 6 | 0.96 ± 0.06 | 0.62 ± 0.05 |
| 7 | 1.06 ± 0.06 | 0.68 ± 0.05 |
| 8 | 1.18 ± 0.09 | 0.80 ± 0.07 |
| 9 | 0.88 ± 0.04 | 0.60 ± 0.04 |
| 10 | 0.69 ± 0.03 | 1.27 ± 0.11 |
| 11 | 0.55 ± 0.02 | 0.84 ± 0.06 |
| 12 | 0.52 ± 0.02 | 0.68 ± 0.04 |
| 13 | 2.11 ± 0.76 | 0.98 ± 0.37 |
| 14 | 1.11 ± 0.24 | 0.49 ± 0.12 |
| 15 | 1.15 ± 0.29 | 0.45 ± 0.12 |

**Updates to Figures.** Aside from the figures discussed above, we regenerate all the model figures with the revised aging-VBS model.

Revised Fig. 5:

[Figure]

Revised Fig. 6:

[Figure]

Revised Fig. S8:

[Figure]

**References**

[revised manuscript text omitted]

---

## Author Comment (AC2)

We appreciate the anonymous reviewers for their thoughtful reviews and comments. We have carefully considered their suggestions and revised the manuscript accordingly. In addition to changes arising from the reviewers, we have made edits to correct figure references in the text and the figures themselves.

Key changes from the previous version are the separation of the gas and particle-phase chemical aging rate coefficients in the aging-VBS model and the addition of model sensitivity evaluations. With the revised model, we regenerate the model figures. We also include an ensemble of optimized parameter sets to provide a range of potential fit values and show the root mean square errors of the models against the data. Accordingly, the text has been updated so $k_{age,particle}$ is not tied to being 10 % of $k_{age,gas}$, including the abstract.

The reviewer comments are in blue, our comments are in black, and modifications to the manuscript are in red. Other edits and corrections not instigated by the reviewers are summarized at the end.

Reviewer # 2

This manuscript investigated the oxidation of D5 siloxane in an oxidation flow reactor (OFR) and the formation of volatile oxidation products, such as formaldehye (HCHO) and formic acid (HCOOH), and secondary organic aerosol (SOSiA). It was found that there was substantial formation of HCHO, HCOOH and SOSiA, highlighting their environmental importance. To reconcile the discrepant SOSiA yields reported in the literature, the study employed a volatility-based multi-generation aging model (VBS) to fit the observed SOSiA in all experiments simultaneously, by tuning the volatility and aging parameters. It was found that the model was able to better capture SOSiA formation from the literature with aging accounted for. This suggested that multi-generational aging may be very important for D5 siloxane oxidation, more so than other systems like monoterpene oxidation, where SOA forms early on. This study is generally well designed, and acceptance is recommended if the comments can be addressed.

We thank the anonymous reviewer for highlighting areas where the manuscript can be improved prior to publication. We updated the manuscript as outlined below.

1. When simulating the OFR experiments, were the OH concentrations corrected for suppression by external reactivity?

We use the $D_5$ measurements from the PTR-MS to derive the $OH_{exp}$ (Eq. 2) during the experiments instead of relying on offline $OH_{exp}$ calibrations. Since the model uses a constant [OH] derived from the empirical $OH_{exp}$, suppression effects are accounted for empirically. While designing the experiments, we mitigated the risk of OH suppression by keeping the $OHR_{ext}$ low. For instance, the highest experimental $[D5]_0$ was 167 ppb, which corresponds to an $OHR_{ext}$ of 8.3 s$^{-1}$, which is within the recommended PAM-OFR conditions by Peng and Jimenez (2020).

Added lines 158-159:

"With these target $[D_5]_0$, we get external OH reactivities ($OHR_{ext}$) of $2.5 - 9.8$ s$^{-1}$ at 298.15 K and 1 atm (Peng and Jimenez, 2020), where $OHR_{ext}$ is the reactivity caused by the injection of $D_5$ into

the PAM-OFR. With these $OHR_{ext}$, we reduce the risk of OH suppression and VOC photolysis (Peng and Jimenez, 2020)."

2. Since HCHO and HCOOH are continuously formed with aging, why quantify their molar yield at zero OH exposure? The yield would change significantly with aging, is that right? Please clarify.

The reviewer is correct that HCHO and HCOOH are likely formed continuously throughout the course of the experiment. We explain in Sect. 3.1.2 that HCHO itself is reactive with OH, while HCOOH is not reactive enough to be appreciably lost by OH within the PAM-OFR residence times. Thus, we expect $Y_{HCHO}$ to decrease with higher $OH_{exp}$, while $Y_{HCOOH}$ is less affected, which is consistent with the measurements.

We fit $\gamma_{HCHO}$ in the limit of $OH_{exp} \rightarrow 0$ to estimate the cumulative yield of HCHO; this value is often used to estimate VOC emissions using satellite retrievals of HCHO (Millet et al., 2006). Mao et al. (2009) also suggest that there are unaccounted for HCHO sources, and we wish to report a rough estimation of the ozone formation potential of $D_5$ through its production of HCHO. We have added the following:

Added lines 334-335:

"This $\gamma_{HCHO}$ is also used to correlate satellite column retrievals of HCHO with VOC emissions (Millet et al., 2006), where an empirical value can be used to constrain uncertainty."

3. In the all the OFR experiments simulated, were seed particles added to promote SOSiA condensation? If not, how would new particle formation and kinetically limited particle growth affect the model predictions? Please include this in the discussion.

We appreciate the reviewer for catching these experimental limitations. We did not use seed aerosol in these experiments and do not have time-series measurements of new particle formation from which we can derive nucleation and growth rates. Instead, we have calculated the condensational sink based on the particle size distribution (Palm et al., 2016) measured out of the PAM-OFR in Sect. S1.3. Based on those calculations, we expect the condensation of LVOC to particles to have been rapid compared to loss to the wall. However, this calculation assumes unity mass accommodation, and particle growth may have been slower if the assumption is incorrect. We have made the following additions to the main text:

Added lines 268-269:

"We also report the condensational sink and condensation lifetimes (Palm et al., 2016) calculated using the particle size distributions in Sect. S1.3."

Added lines 493-496:

"The condensation timescale calculations suggest that the loss of low-volatile species to the wall is small (Sect. S1.3), however, these calculations assume a high mass accommodation coefficient for SOSiA and do not account for particle nucleation. Should particle nucleation be delayed or happen slowly, the gas wall loss may be higher than expected, leading to under quantification of SOSiA."

4. This may be related to 3. I think the authors should make clear in the discussion that this study is focused on reconciling the SOSiA yields by accounting for multi-generational aging only, but aging may be not the only factor affecting the yields, especially in OFR experiments where particle kinetics, phase state etc. can play important roles. Limitations should be acknowledged and if possible, sensitivities should be probed.

We thank the reviewer for pointing out these limitations of the current aging-VBS model. In addition to the model limitations discussed in the supplementary, we have edited this section in the conclusion:

Revised lines 492-506:

> "We also find that the aging-VBS model is sensitive to $k_{age,particle}$ (Fig. S11) and not sensitive to $k_{age,gas}$ (Fig. S12), suggesting that heterogeneous aging should be considered in these models. The condensation timescale calculations suggest that the loss of low-volatile species to the wall is small (Sect. S1.3), however, these calculations assume a high mass accommodation coefficient for SOSiA and do not account for particle nucleation. Should particle nucleation be delayed or happen slowly, the gas wall loss may be higher than expected, leading to under quantification of SOSiA. Furthermore, the aging-VBS model assumes that $k_{age,gas}$ is uniform across products or that chemical aging results in a ten-fold decrease in volatility.

> While the proposed model assumes that the particles are internally well mixed, the high [OH] used in OFRs may induce faster radical reactions and dimerization near the particle surface (Zhao et al., 2019), which affects particle composition and equilibrium timescales. While dimers and oligomers have been found in SOSiA (Wu and Johnston, 2017; Avery et al., 2023; Chen et al., 2023), the model currently does not account for particle-phase oligomer formation. How oligomerization in the $D_5$ + OH SOSiA system evolves the volatility distribution and particle properties is currently not considered in the aging-VBS model. Moreover, high degrees of oxidation should lead to fragmentation and increasing volatility (Isaacman-VanWertz et al. 2018), which is also not considered in the aging-VBS model. Hence, multiphase modeling to evaluate SOSiA chemistry and translate experimental findings to atmospheric conditions remains a direction for future research."

5. The use of "pseudo persistent" in line 68 is somewhat unclear. Please add that D5 siloxane has temporary reservoirs in the atmosphere, if that is right.

We agree with the reviewer that the phrase "pseudo persistent" can be misinterpreted. The phrase comes from Xiang et al. (2021), who used it to mean that the constant emissions of $D_5$ make it persist in the atmosphere. We have corrected the text to prevent misinterpretation:

Revised lines 69-71:

> "Siloxanes have been classified as environmentally persistent or emitted continuously to appear as such (Howard and Muir, 2010; Xiang et al., 2021), while other studies have found that methyl siloxanes are removed on timescales of days to weeks (Graiver et al., 2003; Whelan and Kim, 2021)."

**Additional Manuscript Changes**

**Graphical Abstract:** We have replaced the original figure to the one below, which removes the reference to $RO_2$, whose pathways are not considered in the proposed aging-VBS model.

[Figure]

**Correction to "D" and "T" in the molecular formulas.** Previously, we stated that "D" and "T" refer to units of $(CH_3)_2SiO$ and $CH_3SiO$ respectively. We have corrected explanations of this nomenclature throughout the manuscript for consistency with the literature: "D" and "T" refer to silicon center atoms being bonded with two and three oxygens respectively.

**Changed the title of Sect. 3.2.2 from "Reconciling Literature $Y_{SOSiA}$" to "Consolidating Literature $Y_{SOSiA}$" to prevent misinterpretation.** While the reported SOSiA mass yields vary between papers, they can be consolidated with a single aging-VBS model.

**Added model explanation on how experiment temperature variation was accounted for.** We add the below model detail for replicability.

Added lines 226-228:

"Since the experiments had slight variations in temperature, we correct for temperature impacts on $C^*$ between experiments using the Clausius-Clapeyron equation and an enthalpy of vaporization of 60 kJ mol$^{-1}$., which is that of $D_5$ siloxane (Lei et al., 2010)."

**Corrected condensational sink values in Table S3:** We previously misstated the $CS$ to be in units of m$^{-2}$, while it should be in units of s$^{-1}$ and miscalculated the $\tau_{CS}$ to be too fast. Eqs. S1 and S2 have been corrected, and the values in Table S3 have been revised.

Revised Table S3:

"Table S3. Summary of experiment condensational sinks, LVOC condensation lifetimes, and growth factors calculated with the particle size distribution exiting the PAM-OFR as described in Section S1.3."

| ameters | SOA, LVOC $\kappa$ = 0.13, $M$ = 0.200 kg mol$^{-1}$ | | | SOSiA, $D_5$ $\kappa$ = 0.01, $M$ = 0.370 kg mol$^{-1}$ | | |
|---|---|---|---|---|---|---|
| eriment | $CS$ (s$^{-1}$) | $\tau_{CS}$ (s) | Growth Factor | $CS$ (s$^{-1}$) | $\tau_{CS}$ (s) | Growth Fac |
| 1 | $2.57 \times 10^{-2}$ | 38.8 | 1.02 | $1.88 \times 10^{-2}$ | 53.3 | 1.00 |

| | | | | | | |
|---|---|---|---|---|---|---|
| 2 | $3.99 \times 10^{-2}$ | 25.1 | 1.02 | $2.92 \times 10^{-2}$ | 34.3 | 1.00 |
| 3 | $3.88 \times 10^{-2}$ | 25.76 | 1.02 | $2.84 \times 10^{-2}$ | 35.2 | 1.00 |
| 4 | 0.173 | 5.77 | 1.17 | 0.101 | 9.85 | 1.02 |
| 5 | 0.303 | 3.30 | 1.17 | 0.182 | 5.50 | 1.02 |
| 6 | 0.394 | 2.54 | 1.17 | 0.239 | 4.19 | 1.02 |
| 7 | $6.68 \times 10^{-2}$ | 15.0 | 1.02 | $4.95 \times 10^{-2}$ | 20.2 | 1.00 |
| 8 | 0.138 | 7.27 | 1.02 | 0.104 | 9.63 | 1.00 |
| 9 | 0.250 | 3.99 | 1.02 | 0.192 | 5.20 | 1.00 |
| 10 | 0.338 | 2.95 | 1.12 | 0.217 | 4.61 | 1.01 |
| 11 | 0.522 | 1.92 | 1.12 | 0.342 | 2.93 | 1.01 |
| 12 | 0.913 | 1.09 | 1.12 | 0.605 | 1.65 | 1.01 |
| 13 | $1.25 \times 10^{-2}$ | 80.1 | 1.02 | $9.06 \times 10^{-3}$ | 110 | 1.00 |
| 14 | $1.23 \times 10^{-2}$ | 81.3 | 1.02 | $8.96 \times 10^{-3}$ | 112 | 1.00 |
| 15 | $1.01 \times 10^{-2}$ | 99.3 | 1.02 | $7.35 \times 10^{-3}$ | 136 | 1.00 |

**Corrected $Y_{HCHO}$ error values in Table S8.** An error in the error propagation calculation resulted in the errors being overstated.

Revised Table S8"

"Table S8. Experimental molar yields of HCHO and HCOOH. As these species are formed in the OFR at an unknown point, there may be some loss through oxidation with OH. Consequently, the OHexp determined with D5 may not represent the OHexp these VOP experienced."

| Experiment | ΔHCHO/ΔD$_5$ (ppb/ppb) | ΔHCOOH/ΔD$_5$ (ppb/ppb) |
|---|---|---|
| 1 | 1.79 ± 0.25 | 0.94 ± 0.15 |
| 2 | 1.35 ± 0.15 | 0.69 ± 0.09 |
| 3 | 1.21 ± 0.21 | 0.52 ± 0.09 |
| 4 | 1.52 ± 0.11 | 0.90 ± 0.09 |
| 5 | 1.28 ± 0.11 | 0.83 ± 0.09 |
| 6 | 0.96 ± 0.06 | 0.62 ± 0.05 |
| 7 | 1.06 ± 0.06 | 0.68 ± 0.05 |
| 8 | 1.18 ± 0.09 | 0.80 ± 0.07 |
| 9 | 0.88 ± 0.04 | 0.60 ± 0.04 |
| 10 | 0.69 ± 0.03 | 1.27 ± 0.11 |
| 11 | 0.55 ± 0.02 | 0.84 ± 0.06 |
| 12 | 0.52 ± 0.02 | 0.68 ± 0.04 |

| | | |
|---|---|---|
| 13 | 2.11 ± 0.76 | 0.98 ± 0.37 |
| 14 | 1.11 ± 0.24 | 0.49 ± 0.12 |
| 15 | 1.15 ± 0.29 | 0.45 ± 0.12 |

**Updates to Figures.** Aside from the figures discussed above, we regenerate all the model figures with the revised aging-VBS model.

Revised Fig. 5:

[Figure]

Revised Fig. 6:

[Figure]

Revised Fig. S8:

[Figure]

**References**

Avery, A. M., Alton, M. W., Canagaratna, M. R., Krechmer, J. E., Sueper, D. T., Bhattacharyya, N., Hildebrandt Ruiz, L., Brune, W. H., and Lambe, A. T.: Comparison of the Yield and Chemical Composition of Secondary Organic Aerosol Generated from the OH and Cl Oxidation of Decamethylcyclopentasiloxane, ACS Earth Space Chem, https://doi.org/10.1021/acsearthspacechem.2c00304, 2023.

Chen, Y., Park, Y., Kang, H. G., Jeong, J., and Kim, H.: Chemical characterization and formation of secondary organosiloxane aerosol (SOSiA) from OH oxidation of decamethylcyclopentasiloxane, Environ. Sci.: Atmos., https://doi.org/10.1039/D2EA00161F, 2023.

Graiver, D., Farminer, K. W., and Narayan, R.: A Review of the Fate and Effects of Silicones in the Environment, J Polym Environ, 11, 129–136, https://doi.org/10.1023/A:1026056129717, 2003.

Howard, P. H. and Muir, D. C. G.: Identifying New Persistent and Bioaccumulative Organics Among Chemicals in Commerce, Environ Sci Technol, 44, 2277–2285, https://doi.org/10.1021/es903383a, 2010.

Isaacman-VanWertz, G., Massoli, P., O'Brien, R., Lim, C., Franklin, J. P., Moss, J. A., Hunter, J. F., Nowak, J. B., Canagaratna, M. R., Misztal, P. K., Arata, C., Roscioli, J. R., Herndon, S. T., Onasch, T. B., Lambe, A. T., Jayne, J. T., Su, L., Knopf, D. A., Goldstein, A. H., Worsnop, D. R., and Kroll, J. H.: Chemical evolution of atmospheric organic carbon over multiple generations of oxidation, Nat Chem, 10, 462–468, https://doi.org/10.1038/s41557-018-0002-2, 2018.

Lei, Y. D., Wania, F., and Mathers, D.: Temperature-Dependent Vapor Pressure of Selected Cyclic and Linear Polydimethylsiloxane Oligomers, J Chem Eng Data, 55, 5868–5873, https://doi.org/10.1021/je100835n, 2010.

Millet, D. B., Jacob, D. J., Turquety, S., Hudman, R. C., Wu, S., Fried, A., Walega, J., Heikes, B. G., Blake, D. R., Singh, H. B., Anderson, B. E., and Clarke, A. D.: Formaldehyde distribution over North America: Implications for satellite retrievals of formaldehyde columns and isoprene emission, Journal of Geophysical Research: Atmospheres, 111, https://doi.org/https://doi.org/10.1029/2005JD006853, 2006.

Palm, B. B., Campuzano-Jost, P., Ortega, A. M., Day, D. A., Kaser, L., Jud, W., Karl, T., Hansel, A., Hunter, J. F., Cross, E. S., Kroll, J. H., Peng, Z., Brune, W. H., and Jimenez, J. L.: In situ secondary organic aerosol formation from ambient pine forest air using an oxidation flow reactor, Atmos Chem Phys, 16, 2943–2970, https://doi.org/10.5194/acp-16-2943-2016, 2016.

Peng, Z. and Jimenez, J. L.: Radical chemistry in oxidation flow reactors for atmospheric chemistry research, Chem. Soc. Rev., 49, 2570–2616, https://doi.org/10.1039/C9CS00766K, 2020.

Whelan, M. J. and Kim, J.: Application of multimedia models for understanding the environmental behavior of volatile methylsiloxanes: Fate, transport, and bioaccumulation, Integr Environ Assess Manag, n/a, 1–23, https://doi.org/https://doi.org/10.1002/ieam.4507, 2021.

Wu, Y. and Johnston, M. V: Aerosol Formation from OH Oxidation of the Volatile Cyclic Methyl Siloxane (cVMS) Decamethylcyclopentasiloxane, Environ Sci Technol, 51, 4445–4451, https://doi.org/10.1021/acs.est.7b00655, 2017.

Xiang, X., Liu, N., Xu, L., and Cai, Y.: Review of recent findings on occurrence and fates of siloxanes in environmental compartments, Ecotoxicol Environ Saf, 224, 112631, https://doi.org/https://doi.org/10.1016/j.ecoenv.2021.112631, 2021.

Zhao, Z., Tolentino, R., Lee, J., Vuong, A., Yang, X., and Zhang, H.: Interfacial Dimerization by Organic Radical Reactions during Heterogeneous Oxidative Aging of Oxygenated Organic Aerosols, J Phys Chem A 123, 10782–10792, https://doi.org/10.1021/acs.jpca.9b10779, 2019.